# Multiplayer Federated Learning: Reaching Equilibrium with Less Communication

**TaeHo Yoon**       **Sayantan Choudhury**       **Nicolas Loizou**
Department of Applied Mathematics & Statistics
Mathematical Institute for Data Science
Johns Hopkins University
{tyoon7,schoudh8,nloizou}@jhu.edu

## Abstract

Traditional Federated Learning (FL) approaches assume collaborative clients with aligned objectives working toward a shared global model. However, in many real-world scenarios, clients act as rational players with individual objectives and strategic behavior, a concept that existing FL frameworks are not equipped to adequately address. To bridge this gap, we introduce *Multiplayer Federated Learning (MpFL)*, a novel framework that models the clients in the FL environment as players in a game-theoretic context, aiming to reach an equilibrium. In this scenario, each player tries to optimize their own utility function, which may not align with the collective goal. Within MpFL, we propose *Per-Player Local Stochastic Gradient Descent* (PEARL-SGD), an algorithm in which each player/client performs local updates independently and periodically communicates with other players. We theoretically analyze PEARL-SGD and prove that it reaches a neighborhood of equilibrium with less communication in the stochastic setting compared to its non-local counterpart. Finally, we verify our theory through numerical experiments.

## 1   Introduction

Federated Learning (FL) has emerged as a powerful collaborative learning paradigm where multiple clients jointly train a machine learning model without sharing their local data. In the classical FL setting, a central server coordinates multiple clients (e.g., mobile or edge devices) to collaboratively learn a shared global model without exchanging their own training data [59, 66, 94, 77]. In this scenario, each client performs local computations on its private data and periodically communicates model updates to the server, which aggregates them to update the global model. This collaborative approach has been successfully applied in various domains, including natural language processing [83, 52], computer vision [84, 76], and healthcare [5, 135].

Despite their success, traditional FL frameworks rely on the key assumption that all participants are fully cooperative and share aligned objectives, collectively working to optimize the performance of a shared global model (e.g., minimizing the average of individual loss functions). This assumption overlooks situations where participants have individual objectives or competitive interests that may not align with the collective goal. A variety of such scenarios have been extensively considered in the game theory literature, including Cournot competition in economics [2], optical networks [108], electricity markets [116], energy consumption control in smart grid [141], or mobile robot control [60]. These applications have yet to be associated with FL, presenting an underexplored opportunity to bridge game theory and FL for more robust and realistic frameworks.

To address these limitations of classical FL approaches, we propose a novel framework, *Multiplayer Federated Learning (MpFL)*, which models the FL process as a game among rational players with individual utility functions. In MpFL, each participant is considered a player who aims to optimize

their own objective while interacting strategically with other players in the network via a central server. This game-theoretic perspective acknowledges that participants may act in their self-interest, have conflicting goals, or be unwilling to fully cooperate. By incorporating these dynamics, MpFL provides a more realistic and flexible foundation for FL in competitive and heterogeneous environments.

In the literature, there are multiple strategies that aim to incorporate personalization into classical FL, including multi-task learning [124, 98], transfer learning [64], and mixing of the local and global models [49, 50], to name a few. However, to the best of our knowledge, none of them can formulate the behavior of the clients/players in a non-cooperative environment. This gap is precisely what Multiplayer Federated Learning (MpFL) aims to address.

## 1.1 Main contributions

- **Introducing Multiplayer Federated Learning.** We develop a novel framework of *Multiplayer Federated Learning (MpFL)*, which models the FL process as a game among rational players with individual utility functions. In MpFL, each client within the FL environment is viewed as a player of the game, and their local models are viewed as their actions. Each player constantly adjusts their model (action) to optimize their own objective function, and the MpFL framework aims for each player to reach to a Nash equilibrium by collaboratively training their model under the orchestration of a central server (e.g., service provider), while keeping the training data decentralized. That is, MpFL extends the scope of FL to scenarios where clients are allowed to have more general, diversified, possibly competing objectives.

- **Design and analysis of Per-Player Local SGD.** To handle the Multiplayer Federated Learning framework, we introduce *Per-Player Local SGD* (PEARL-SGD), a new algorithm inspired by the stochastic gradient descent ascent method in minimax optimization, that is able to handle the competitive nature of the players/clients. In PEARL-SGD, each player performs local SGD steps independently on their own actions/strategies (keeping the strategies of the other players fixed), and the udpated actions/models are periodically communicated with the other players of the network via a central server.

- **Convergence guarantees for PEARL-SGD on heterogeneous data.** We provide tight convergence guarantees for PEARL-SGD, in both deterministic and stochastic regimes with heterogeneous data (see Table 1 for a summary of our results).

  - **Deterministic setting:** For the full-batch (deterministic) variant of PEARL-SGD, we prove that under suitable assumptions, PEARL-SGD converges linearly to an equilibrium for any communication period $\tau > 1$, provided that the constant step-size $\gamma$ is sufficiently small (see Theorem 3.3).

  - **Stochastic setting:** In its more general version, PEARL-SGD assumes that each player uses an unbiased estimator of its gradient in the update rule. For this setting, we provide two Theorems based on two different step-size choices:

    * *Constant step-size:* We show that under the same assumptions as in the deterministic case, PEARL-SGD converges linearly to a neighborhood of equilibrium (see Theorem 3.4). In Corollary 3.5, we show that with appropriate step-size depending on the total number of local SGD iterations $T$, PEARL-SGD achieves $\tilde{\mathcal{O}}(1/T)$ convergence rate with improved communication complexity when $T$ is sufficiently large.

    * *Decreasing step-size rule:* We prove that PEARL-SGD converges to an exact equilibrium (without neighborhood of convergence) with sublinear convergence (see Theorem 3.6). In this scenario, the asymptotic rate and communication complexity are essentially the same as in Corollary 3.5, but this result does not require the step-sizes to depend on $T$.

- **Numerical Evaluation:** We provide numerical experiments verifying our theoretical results and show the benefits in terms of communications of PEARL-SGD over its non-local counterpart in the MpFL settings.

## 2 Multiplayer Federated Learning: Definition and Related Work

In this section, we introduce the Multiplayer Federated Learning (MpFL) framework and explain its main differences compared to the classical FL [59], federated minimax optimization [25, 122, 145] and personalized FL [34, 129].

Table 1: Summary of theoretical results for PEARL-SGD. Theorem 3.3 considers the full-batch (deterministic) scenario. Theorem 3.4 and Theorem 3.6 both considers the general stochastic case. These results differ in the step-size choice; the former uses a constant step-size, while the latter uses decreasing step-sizes. In the *Convergence* column, "Linear" and "Sublinear" indicates the convergence rate, "Exact" refers to convergence to an equilibrium, and "Neighborhood" refers to convergence to a neighborhood of an equilibrium.

| *Theorem* | *Setting* | *Step-size* | *Convergence* |
|-----------|-----------|-------------|---------------|
| Theorem 3.3 | Deterministic | Constant | Linear+Exact |
| Theorem 3.4 | Stochastic | Constant | Linear+Neighborhood |
| Theorem 3.6 | Stochastic | Decreasing | Sublinear+Exact |

## 2.1 Definition of MpFL

*Multiplayer Federated Learning (MpFL)* is a machine learning setting that combines the benefits of a game-theoretic formulation with classical federated learning. In this setting, the problem is an $n$-player game in which multiple players/clients (e.g. mobile devices or whole organizations) communicate with each other via a central server (e.g. service provider) to reach equilibrium. That is, reach a set of strategies where no player can unilaterally deviate from their strategy to achieve a better payoff, given the strategies chosen by all other players.

In classical $n$-player games, communication between players was assumed to be cheap, easy, and straightforward, mainly because all players were in close proximity and had direct access to one another. This assumption made communication an insignificant concern in typical game theory analysis. However, with the advent of new large-scale machine learning applications, this is no longer the case. Communication between players can be expensive and challenging, especially in distributed systems where the clients/players are geographically dispersed or operate under communication constraints. Addressing communication costs and designing communication-efficient algorithms for $n$-player games have become increasingly important, and this is precisely the challenge that Multiplayer Federated Learning aims to address.

**Equilibrium in $n$-player game.** Let $x^i \in \mathbb{R}^{d_i}$ denote the action of player $i = 1, \ldots, n$ and let $\mathbf{x} = (x^1, \ldots, x^n) \in \mathbb{R}^D = \mathbb{R}^{d_1 + \cdots + d_n}$ be the joint action/strategy vector of all players. Let $f_i(x^1, \ldots, x^n) \colon \mathbb{R}^{d_1 + \cdots + d_n} \to \mathbb{R}$ be the function of the player $i$ (which player $i$ prefers to minimize in $x^i$) and let $x^{-i} = (x^1, \ldots, x^{i-1}, x^{i+1}, \ldots, x^n) \in \mathbb{R}^{D-d_i}$ be the vector containing all players' actions except that of player $i$. With this notation in place, the goal of an $n$-player game is to find an *equilibrium*, a joint action $\mathbf{x}_\star = (x_\star^1, \ldots, x_\star^n) \in \mathbb{R}^D$, formally expressed as:

$$\underset{\mathbf{x}_\star = (x_\star^1, \ldots, x_\star^n) \in \mathbb{R}^D}{\text{find}} \quad f_i(x_\star^i; x_\star^{-i}) \leq f_i(x^i; x_\star^{-i}), \quad \forall x^i \in \mathbb{R}^{d_i}, \quad \forall i \in [n], \tag{1}$$

where $f_i(x^i; x^{-i}) = f_i(x^1, \ldots, x^n)$.

**MpFL.** As mentioned above, in the setting of interest of this work, we focus on an $n$-player game in which multiple players communicate via a central server to reach an equilibrium. In this setting, each player of the $n$-player game represents a client to the system (see Figure 1). Mathematically, the problem is formulated as solving (1) with

$$f_i(x^1, \ldots, x^n) = \mathbb{E}_{\xi^i \sim \mathcal{D}_i} \left[ f_{i, \xi^i}(x^1, \ldots, x^n) \right].$$

Here $\mathcal{D}_i$ denotes the data distribution of the $i$-th player, $f_{i, \xi^i}$ is the loss of the $i$-th player for a data point $\xi^i$ sampled from $\mathcal{D}_i$.

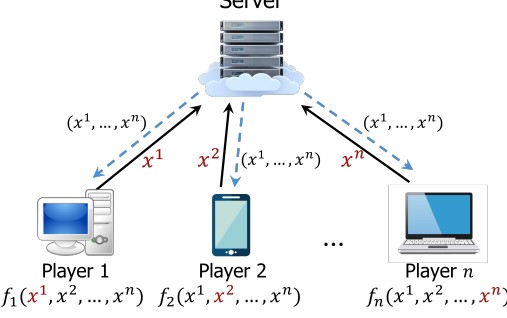

Figure 1: Illustration of MpFL for heterogeneous functions $f_i$. The goal is for each player to reach the equilibrium $\mathbf{x}_\star = (x_\star^1, \ldots, x_\star^n)$ (see (1)) with as little communication as possible.

In our proposed FL environment, each client/player uses the strategies of all players to execute local updates. In particular, each player keeps the other players' strategies fixed and updates their own value, which is later shared with the master server, which concatenates all new strategies and sends them back to all players. Later, in Section 3 we introduce and analyze Algorithm 1, named *Per-Player Local SGD* (PEARL-SGD), which formalizes the above setting.

Similarly to classical FL, our setting allows for *heterogeneous (non-iid) data* as we have no restrictive assumption on the data distribution $\mathcal{D}_i$ or the similarity between the functions of the players.

**Assumptions on multiplayer game.** Let us present the main assumptions on the functions of the multiplayer game, which we later use to provide the convergence analysis for PEARL-SGD— objective function $f_i$ of each player $i \in [n]$ is convex and smooth.

Throughout this work, we denote the partial derivative of $f_i$ with respect to $x^i$ as $\nabla_{x^i} f_i(x^1, \ldots, x^n) = \nabla f_i(x^i; x^{-i})$. This convention allows us to remove the cumbersome subscript $x^i$ from the $\nabla$ notation, with the understanding that we only differentiate $f_i$ with respect to $x^i$ but never with $x^{-i}$.

**Assumption 2.1** (*Convex (CVX)*). For $i \in [n]$, for any $x^{-i} \in \mathbb{R}^{D-d_i}$, the local function $f_i(\cdot; x^{-i}) \colon \mathbb{R}^{d_i} \to \mathbb{R}$ is convex. That is, for any $x^i, y^i \in \mathbb{R}^{d_i}$ and $x^{-i} \in \mathbb{R}^{D-d_i}$,

$$f_i(y^i; x^{-i}) \geq f_i(x^i; x^{-i}) + \left\langle \nabla f_i(x^i; x^{-i}), y^i - x^i \right\rangle$$

**Assumption 2.2** (*Smoothness (SM)*). For $i \in [n]$, for any $x^{-i} \in \mathbb{R}^{D-d_i}$, the local function $f_i(\cdot; x^{-i}) \colon \mathbb{R}^{d_i} \to \mathbb{R}$ is $L_i$-smooth. That is, for any $x^i, y^i \in \mathbb{R}^{d_i}$ and $x^{-i} \in \mathbb{R}^{D-d_i}$,

$$\left\| \nabla f_i(x^i; x^{-i}) - \nabla f_i(y^i; x^{-i}) \right\| \leq L_i \left\| x^i - y^i \right\|.$$

As in (1), in the stochastic regime of MpFL we have $f_i(x^1, \ldots, x^n) = \mathbb{E}_{\xi^i \sim \mathcal{D}_i} \left[ f_{i,\xi^i}(x^1, \ldots, x^n) \right]$. To obtain convergence guarantees for PEARL-SGD in this scenario, we need the following assumption of bounded variance of the gradient oracle, commonly used in stochastic optimization.

**Assumption 2.3** (*Bounded Variance (BV)*). Let $\sigma_i \geq 0$, $\forall i \in [n]$. For each $i = 1, \ldots, n$,

$$\mathbb{E}_{\xi^i \sim \mathcal{D}_i} \left[ \left\| \nabla f_{i,\xi^i}(x^i; x^{-i}) - \nabla f_i(x^i; x^{-i}) \right\|^2 \right] \leq \sigma_i^2, \quad \forall x^i \in \mathbb{R}^{d_i}, x^{-i} \in \mathbb{R}^{D-d_i}.$$

### 2.2 Comparison with closely related FL frameworks

Having presented the MpFL setting, let us now provide a concise survey of the most closely related setups: classical FL, federated minimax optimization and personalized FL. We compare each of them with MpFL, and explain in detail in Appendix B *why existing FL algorithms are not applicable to MpFL*. An additional list of related work, explaining the literatures on game theory, distributed Nash equilibrium search, learning in games and the usage of game theory for modeling clients' social behavior in FL, and how MpFL is distinguished from them, is provided in Appendix A.

**Federated learning (FL).** The basic formulation for classical FL is [59]: $\underset{x \in \mathbb{R}^d}{\text{minimize}} \ f(x) = \frac{1}{n} \sum_{i=1}^{n} f_i(x)$. Here, $x \in \mathbb{R}^d$ represents the global model parameter, $f_i(x) = \mathbb{E}_{\xi^i \sim \mathcal{D}_i}[F_i(x, \xi^i)]$ denotes the local objective function at client $i$, and $\mathcal{D}_i$ denotes the data distribution of client $i$. The local loss functions $F_i(x, \xi^i)$ are often of the same form across all clients, but the data distribution $\mathcal{D}_i$ generally varies, capturing data heterogeneity. The foundational communication-efficient algorithm for this setup is FedAvg (Local SGD), proposed and popularized by [93]. Despite its simplicity, Local SGD has shown empirical success in terms of convergence speed and communication cost. Many works provided theoretical explanation for this performance [127, 29, 128, 63].

Note that in these works on classical FL, clients work in a fully cooperative manner to find $x_\star = \operatorname{argmin}_{x \in \mathbb{R}^d} f(x)$, unlike in our proposed MpFL, where the clients serve as players of the game and seek an equilibrium among possibly competing (non-cooperative) objectives.

**Federated minimax optimization.** This is a more recent, federated extension of minimax optimization appearing in many ML applications. There the problem is: $\displaystyle\min_{x\in\mathbb{R}^{d_x}}\max_{y\in\mathbb{R}^{d_y}}\mathcal{L}(x,y)=\frac{1}{n}\sum_{i=1}^{n}\mathcal{L}_i(x,y)$. Here $n$ is the number of clients and $\mathcal{L}_i$ represents the local loss function at client $i$ that depends on both $x$ and $y$. It is defined as $\mathcal{L}_i(x,y)=\mathbb{E}_{\xi^i\sim\mathcal{D}_i}[\phi_i(x,y,\xi)]$, where $\phi_i(x,y,\xi)$ denotes the loss for the data point $\xi$, sampled from the local data distribution $\mathcal{D}_i$ at client $i$. The extension of Local SGD for solving this problem are Local Stochastic Gradient Descent-Ascent (SGDA) [25, 122] or Local Stochastic Extragradient (SEG) [8, 9] algorithms. More recently there was also an approach based on primal-dual updates [20].

While this line of work also studied federated learning in the context of minimax optimization and games, it is totally different from MpFL. The setup assumes that *each FL client adjusts the actions of both players $x$ and $y$ of the minimax game*, and does not take the *multiplayer* aspect into account. In contrast, in MpFL, we assume that each client $i$ is a player of a large-scale multiplayer game who *only adjusts their own action $x^i$* in the interest of optimizing their own objective $f_i$. In our work, we design the novel PEARL-SGD algorithm suitable for MpFL, as the existing Local SGDA and Local SEG methods cannot handle the setup.

**Personalized federated learning.** In personalized FL [34, 129, 50, 49, 26, 130], clients aim to learn models tailored to each local data distribution, while generalizing well on all clients' data [51]. One formulation of personalized FL is $\displaystyle\min_{\mathbf{x}=(x^1,\ldots,x^n)\in\mathbb{R}^{nd}}\frac{1}{n}\sum_{i=1}^{n}h_i(x^i)+\phi(x^1,\ldots,x^n)$; in [50, 51], for example, $\phi$ is taken as the model consensus regularizer $\phi(x^1,\ldots,x^n)=\frac{\lambda}{2n}\sum_{i=1}^{n}\left\|x^i-\overline{x}\right\|^2$. Given that each $h_i$ and $\phi$ are convex, its first-order optimality condition is equivalent to the equilibrium condition for the $n$-player game where players' objectives are $f_i(x^i;x^{-i})=h_i(x^i)+n\phi(x^1,\ldots,x^n)$. Such formulation of personalized FL, therefore, is an instance of MpFL. On the other hand, there exist personalized FL approaches that are not modeled as a direct subclass of MpFL—e.g., [34, 129] proposed to maintain a separate global model $w$ along with local models $x^i$. Still, it is clear that MpFL is closely related to personalized FL. Importantly, however, the purpose of MpFL is not limited to having personalized models suitable for local data distribution, and encompasses settings where $x^1,\ldots,x^n$ differ in dimensionality or structure.

## 3 PEARL-SGD: Algorithm and Convergence Guarantees

In this section, we introduce and analyze Algorithm 1, named *Per-Player Local SGD* (PEARL-SGD), which is suitable for the MpFL setting we described in Section 2.

### 3.1 Algorithm and assumptions

In PEARL-SGD, clients/players of the game run SGD independently in parallel to update their strategy (keeping the strategies $x^{-i}$ of the other players fixed), which are communicated to other players only once in a while. In more detail, in every round of PEARL-SGD, each player $i\in[n]$ runs $\tau$ iterations of stochastic gradient descent (SGD) with respect to $f_i(\cdot,x^{-i})$, having $x^{-i}$ fixed to be the information of the other players' actions obtained from the previous synchronization step. Once each player completes $\tau$ iterations of SGD (local updates), a synchronization occurs; the central server collects the actions of all players, and then the concatenation of all updated strategies/actions is distributed to all clients/players.

---

**Algorithm 1** Per-Player Local SGD (PEARL-SGD)

**Input:** Step-sizes $\gamma_k>0$, Synchronization interval $\tau\geq1$, Number of rounds $R\geq1$

    **for** $p=0,\ldots,R-1$ **do**
        Master collects $x_{\tau p}^i$ from players $i\in[n]$
        Master distributes $\mathbf{x}_{\tau p}$ back to players
        **for each players** $i=1,\ldots,n$ **in parallel do**
            **for** $k=\tau p,\ldots,\tau(p+1)-1$ **do**
                Draw $\xi_k^i\sim\mathcal{D}_i$
                $g_k^i\leftarrow\nabla f_{i,\xi_k^i}(x_k^i;x_{\tau p}^{-i})$
                $x_{k+1}^i\leftarrow x_k^i-\gamma_k g_k^i$
            **end for**
        **end for**
    **end for**

**Output:** $\mathbf{x}_{\tau R}\in\mathbb{R}^D$

---

We emphasize that PEARL-SGD and its convergence guarantees hold without any assumption on players' data distributions $\mathcal{D}_i$. That is, functions $f_i$ can be very different between players, and *the setting is fully heterogeneous.*

Let us note that the synchronization step in PEARL-SGD involves transferring a $D = (d_1 + \cdots + d_n)$-dimensional vector $\mathbf{x}_{\tau p}$ from the master server to the players. This is an important difference compared to the classical FL (minimization problem), where the dimension of the communication vectors is the same from client to master and from master to client, and it does not scale with $n$. While PEARL-SGD aims to reduce this overhead compared to its distributed variant ($\tau = 1$) by communicating less frequently (with $\tau > 1$), the high complexity of the synchronization step makes MpFL (and distributed $n$-player games in general) more suitable for cross-silo FL setups with relatively small number of organizations and more reliable communication bandwidths. We expect that the potentially expensive communication of high-dimensional vectors $\mathbf{x}_{\tau p}$ could be addressed by incorporating additional techniques such as gradient compression [4, 10]. This is an orthogonal approach to our proposed local methods, and we leave it for future work.

**Assumptions on the joint gradient operator.** We require some definitions and additional assumptions in order to carry out the theory. Define the joint gradient operator $\mathbb{F}: \mathbb{R}^D \to \mathbb{R}^D$ as

$$\mathbb{F}(\mathbf{x}) = \left( \nabla f_1(x^1; x^{-1}), \ldots, \nabla f_n(x^n; x^{-n}) \right).$$

**Assumption 3.1** (*Quasi-strong monotonicity (QSM)*). There exists $\mathbf{x}_\star = (x_\star^1, \ldots, x_\star^n) \in \mathbb{R}^D$, an equilibrium where $\mathbb{F}(\mathbf{x}_\star) = 0$ and $\mu > 0$ such that $\forall \mathbf{x} \in \mathbb{R}^D$, $\langle \mathbb{F}(\mathbf{x}), \mathbf{x} - \mathbf{x}_\star \rangle \geq \mu \|\mathbf{x} - \mathbf{x}_\star\|^2$.

*(QSM)* is a concept extending quasi-strong convexity [46, 45] to the context of variational inequality problems (VIPs). This condition has been referred to as different names in the literature, such as strong coherent VIPs [126], VIPs with strong stability condition [95], or the strong Minty variational inequality [28]. It generalizes strong monotonicity, capturing some non-monotone problems. In [89], it was proposed and utilized as an assumption ensuring the convergence of SGDA dynamics in minimax games without the well-known issues of cycling or diverging [97, 22]. Later, it was also used in the analysis of stochastic extragradient method [42] and its single-call variants (optimistic and past stochastic etragradient) [19].

**Assumption 3.2** (*Star-cocoercivity (SCO)*). $\mathbb{F}$ is $\frac{1}{\ell}$-star-cocoercive, i.e., there is $\ell > 0$ such that for any $\mathbf{x} \in \mathbb{R}^D$, $\langle \mathbb{F}(\mathbf{x}), \mathbf{x} - \mathbf{x}_\star \rangle \geq \frac{1}{\ell} \|\mathbb{F}(\mathbf{x})\|^2$.

*(SCO)* generalizes the class of coercive operators and, interestingly, can hold for non-Lipschitz operators [89]. This has also been taken as minimal assumption for SGDA analysis in prior work [10]. Note that *(QSM)* and *(SCO)* together imply $\mu \|\mathbf{x} - \mathbf{x}_\star\| \leq \|\mathbb{F}(\mathbf{x})\| \leq \ell \|\mathbf{x} - \mathbf{x}_\star\|$ for any $\mathbf{x} \in \mathbb{R}^D$, which implies $\mu \leq \ell$. We call $\kappa = \ell/\mu \geq 1$ the *condition number* of the problem. In Appendix F, we provide a detailed discussion on the set of our theoretical assumptions and explain connections to other commonly assumed properties in the literature such as cocoercivity, Lipschitzness and monotonicity.

### 3.2 Convergence of PEARL-SGD: Deterministic setup

First, we provide the convergence result for PEARL-SGD with constant step-size $\gamma_k \equiv \gamma$ in the full-batch (deterministic) scenario, where there is no noise in the gradient computation. While this is directly recovered as a special case of Theorem 3.4 with $\sigma_i = 0$, we state it separately as the deterministic case provides several points of discussion that are worth emphasizing on their own.

**Theorem 3.3.** Assume *(CVX)*, *(SM)*, *(QSM)* and *(SCO)*. Let $L_{\max} = \max\{L_1, \ldots, L_n\}$, $\kappa = \ell/\mu$ and $0 < \gamma_k \equiv \gamma \leq \frac{1}{\ell\tau + 2(\tau - 1) L_{\max} \sqrt{\kappa}}$. Then the Deterministic PEARL-SGD (Algorithm 1 with full-batch) converges with the rate

$$\|\mathbf{x}_{\tau R} - \mathbf{x}_\star\|^2 \leq (1 - \gamma\tau\mu\zeta)^R \|\mathbf{x}_0 - \mathbf{x}_\star\|^2$$

where $\zeta = 2 - \gamma\ell\tau - 2(\tau - 1)\gamma L_{\max}\sqrt{\kappa/3} > 0$ (by the choice of $\gamma$).

Theorem 3.3 shows that deterministic PEARL-SGD converges linearly to an equilibrium. This is in contrast with the case of local gradient descent in FL setup with heterogeneous data, where convergence is only to a neighborhood of the optimum even in the absence of noise [63], unless further correction mechanism is used [100]. This is because the classical FL problem is modeled as finite sum minimization, whereas our MpFL is modeled as a game, for which the existence of a variationally stable equilibrium is a standard assumption for convergence analysis [14, 95, 89, 96]. In particular, when $\tau = 1$, the step-size constraint and the convergence rate of Theorem 3.3 reduces to the analysis of gradient descent-ascent (GDA) under the *(QSM)* and *(SCO)* assumptions from [89], showing that our analysis is tight and consistent with the existing literature.

**Player drift and step-size constraint.** If $\gamma$ does not appropriately scale down with $\tau$, then at each round, players' actions (SGD iterates) converge to minimizers of local functions. We call this phenomenon *player drift*, analogous to client drift in classical FL [61], enforcing the $\mathcal{O}(1/\tau)$ step-size. In our setting, note that the local minimizers $x_\star^i(x_{\tau p}^{-i}) := \operatorname{argmin}_{x^i \in \mathbb{R}^{d_i}} f_i(x^i; x_{\tau p}^{-i})$ depend on other players' strategies $x_{\tau p}^{-i}$. Due to this dependence, under extreme player drift, PEARL-SGD may display undesirable dynamics such as diverging to infinity (which can be checked with simple examples such as the two-player quadratic minimax game $\min_{u \in \mathbb{R}} \max_{v \in \mathbb{R}} \frac{\mu}{2} u^2 + uv - \frac{\mu}{2} v^2$ with $\mu < 1$). As these features are not typically observed in client drift in classical FL, player drift represents a distinct phenomenon despite some conceptual similarities. Therefore, we consider understanding and mitigating player drift an intriguing direction for future work in MpFL, which may necessitate novel insights that differ from existing approaches to client drift [61, 100].

With the step-size constrained to $\gamma = \mathcal{O}(1/\tau)$, communication reduction by PEARL-SGD is not observed in the deterministic setting (Theorem 3.3) but is achieved in the stochastic setting—see Section 3.3. A concurrent work [148] proposes the *Decoupled SGD* algorithm, which coincides with our PEARL-SGD, and shows that it can be communication-efficient even in the deterministic setup under an additional assumption of *weakly coupled* games (with slightly different main assumptions). We provide a more complete discussion of this in Appendix A.

### 3.3 Convergence of PEARL-SGD: Stochastic setup

We now discuss the convergence of PEARL-SGD with stochastic gradients. We first present the convergence of PEARL-SGD to a neighborhood of the equilibrium $\mathbf{x}_\star$ given constant step-sizes $\gamma_k \equiv \gamma$, and then discuss the communication complexity gain we achieve. Then we present the convergence result using a decreasing step-size selection, showing exact sublinear convergence to $\mathbf{x}_\star$ (rather than its neighborhood). We provide the outline and details of the proofs in Appendix C.

**Theorem 3.4.** Assume *(CVX)*, *(SM)*, *(BV)*, *(QSM)* and *(SCO)* hold. Let $0 < \gamma_k \equiv \gamma \le \frac{1}{\ell\tau + 2(\tau-1)L_{\max}\sqrt{\kappa}}$ and denote $q = L_{\max}/\sqrt{\ell\mu}$. Then PEARL-SGD exhibits the rate:

$$\mathbb{E}\left[\|\mathbf{x}_{\tau R} - \mathbf{x}_\star\|^2\right] \le (1 - \gamma\tau\mu\zeta)^R \|\mathbf{x}_0 - \mathbf{x}_\star\|^2 + \left(1 + (\tau-1)\left((4 + \sqrt{3}q)\gamma\tau L_{\max} + \frac{q}{2\tau}\right)\right)\frac{\gamma\sigma^2}{\mu\zeta}.$$

where $\sigma^2 = \sum_{i=1}^n \sigma_i^2$ and $\zeta = 2 - \gamma\ell\tau - 2(\tau-1)\gamma L_{\max}\sqrt{\kappa/3} > 0$ by the choice of $\gamma$.

When $\tau = 1$, with $\gamma \le 1/\ell$, the above rate becomes $\mathbb{E}\left[\|\mathbf{x}_R - \mathbf{x}_\star\|^2\right] \le (1 - \gamma\mu)^R \|\mathbf{x}_0 - \mathbf{x}_\star\|^2 + \gamma\sigma^2/\mu$, which is consistent with the classical analysis of the stochastic gradient descent-ascent (SGDA). In the result, note that $\sigma^2$ is the sum of $\sigma_i^2$'s, the (upper bounds on) playerwise gradient variances $(\sigma_i^2 \ge \mathbb{E}_{\xi^i \sim \mathcal{D}_i}\left[\left\|\nabla f_{i,\xi^i}(x^i; x^{-i}) - \nabla f_i(x^i; x^{-i})\right\|^2\right])$. Hence, $\sigma^2$ represents the upper bound on the variance in estimating the joint gradient operator $\mathbb{F}(\cdot)$.

**Remark.** If we use the largest possible step-size $\gamma = \frac{1}{\ell\tau + 2(\tau-1)L_{\max}\sqrt{\kappa}}$ allowed in Theorem 3.4, then the right-hand side of the bound does not scale down indefinitely with $\tau$. In fact, with this choice of $\gamma$, one can expect the communication gain by a factor of approximately $\frac{L_{\max}}{\ell}$ (when $L_{\max} \ll \ell$). More precisely, suppose $q \le 1$ (equivalently $L_{\max} \le \sqrt{\ell\mu}$—refer to Appendix F for the explanation

that this is a common parameter regime). Then we have $\gamma = \Theta\left(\frac{1}{\ell\tau}\right)$ and

$$\mathbb{E}\left[\|\mathbf{x}_{\tau R} - \mathbf{x}_\star\|^2\right] \leq (1 - \gamma\tau\mu)^R \|\mathbf{x}_0 - \mathbf{x}_\star\|^2 + \mathcal{O}\left(\frac{1}{\tau} + \frac{L_{\max}}{\ell}\right)\frac{\sigma^2}{\ell\mu}.$$

The first linear convergence term is essentially unaffected by $\tau$, as the effect of using smaller $\gamma = \Theta\left(\frac{1}{\ell\tau}\right)$ is canceled out by the factor $\tau$ within $(1 - \gamma\tau\mu)^R$. In the second term (which is usually dominant), we see that the size of the convergence neighborhood is reduced by the factor $\mathcal{O}\left(\frac{1}{\tau} + \frac{L_{\max}}{\ell}\right) = \mathcal{O}\left(\frac{1}{\tau} + \frac{1}{\sqrt{\kappa}}\right)$. Therefore, we see that with $\tau = \Omega(\sqrt{\kappa})$, PEARL-SGD reaches about $\sqrt{\kappa}$ times smaller neighborhood within the same number of communication rounds $R$ (compared to the case $\tau = 1$).

In Corollary 3.5, we analyze the convergence and communication gain of PEARL-SGD in the regime where the total number of iterations $T = \tau R$ is large, using a step-size depending on $T$.

**Corollary 3.5.** Suppose the assumptions of Theorem 3.4 hold, and let $\tau \geq 1$ be fixed. Let $q = L_{\max}/\sqrt{\ell\mu}$. Then PEARL-SGD with $\gamma_k \equiv \gamma = \frac{1}{\mu\eta(1+2q)}$ exhibits the rate

$$\mathbb{E}\left[\|\mathbf{x}_T - \mathbf{x}_\star\|^2\right] = \tilde{\mathcal{O}}\left(\frac{(1+q)^2 \|\mathbf{x}_0 - \mathbf{x}_\star\|^2}{T^2} + \frac{(1+q)\sigma^2}{\mu^2 T} + \frac{(1+q)\tau^2 L_{\max}\sigma^2}{\mu^3 T^2}\right)$$

where $\eta$ is selected so that $T = 2(1 + 2q)\eta\log\eta$, provided that $T$ is large enough so that $\eta > \kappa\tau$. Here the $\tilde{\mathcal{O}}$-notation hides polylogarithmic terms in $T$ and constant factors.

**Reduction of communication complexity.** Note that the $\tilde{\mathcal{O}}\left(\frac{(1+q)^2\|\mathbf{x}_0-\mathbf{x}_\star\|^2}{T^2}\right)$ term decays fast in Corollary 3.5 (as $T$ grows) and the terms proportional to $\sigma^2$ become dominant. The order of convergence is not slower than the $\tilde{\mathcal{O}}(1/T)$ rate of the fully communicating case $\tau = 1$, provided that $\tau^2 L_{\max}\sigma^2/\mu^3 T^2 = \mathcal{O}(\sigma^2/\mu^2 T) \iff \tau = \mathcal{O}\left(\sqrt{\mu T/L_{\max}}\right)$. Therefore, as long as we select $\tau = \mathcal{O}\left(\sqrt{\mu T/L_{\max}}\right)$, in PEARL-SGD the communication cost is reduced by the factor of $\tau$ (because the total number of communications is $T/\tau$). With the largest possible $\tau$, the resulting communication complexity is $T/\tau = \Theta\left(\sqrt{TL_{\max}/\mu}\right) = \Theta\left(\sqrt{T}\right)$.

**Convergence to equilibrium via decreasing step-sizes.** We conclude the section with convergence result for PEARL-SGD using a decreasing step-size selection. While showing a similar convergence rate in terms of $T$ as in Corollary 3.5, Theorem 3.6 has the advantage of not requiring to fix $T$ in advance to determine the step-sizes.

**Theorem 3.6.** Under the assumptions of Theorem 3.4, let $q = L_{\max}/\sqrt{\ell\mu}$, and choose the step-sizes $\gamma_k = \begin{cases} \frac{1}{\ell\tau(1+2q)} & \text{if } p < 2(1+2q)\kappa \\ \frac{1}{\tau\mu}\frac{2p+1}{(p+1)^2} & \text{if } p \geq 2(1+2q)\kappa \end{cases}$ for $\tau p \leq k \leq \tau(p+1) - 1, p = 0, \ldots, R - 1$. Then PEARL-SGD converges with the rate

$$\mathbb{E}\left[\|\mathbf{x}_T - \mathbf{x}_\star\|^2\right] \leq \frac{4(1+2q)^2\kappa^2\tau^2 \|\mathbf{x}_0 - \mathbf{x}_\star\|^2}{eT^2} + \frac{4(1+q)\sigma^2}{\mu^2 T}$$

$$+ \frac{4(1+2q)^2\kappa\tau\sigma^2}{\mu^2 T^2}\left(1 + \frac{2\tau}{\sqrt{\kappa}}\right) + \frac{32(1+q)\tau^2 L_{\max}\sigma^2\log T}{\mu^3 T^2}$$

where $T = \tau R$ is the total number of iterations.

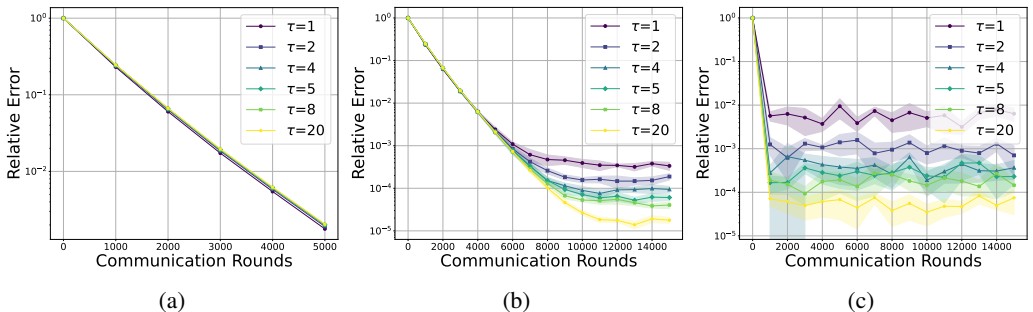

Figure 2: Performance plots for PEARL-SGD. Figures 2a (deterministic) and 2b (stochastic) show the relative error $\frac{\|\mathbf{x}_k - \mathbf{x}_\star\|^2}{\|\mathbf{x}_0 - \mathbf{x}_\star\|^2}$ on the $n$-player game defined by (2) with different values of $\tau$, using theoretical step-sizes. (We provide additional experiments for this $n$-player game setup in Appendix E.) Figure 2c shows the relative error in the (stochastic) mobile robot control setup (3) for distinct values of $\tau$.

## 4 Numerical Experiments

In this section, we conduct experiments to assess the empirical performance of PEARL-SGD and verify our theory. We focus on two setups: a multiplayer game with quadratic objectives, and a distributed mobile robot control problem. Details of experiments are provided in Appendix D.

### 4.1 Quadratic $n$-player game

We consider an $n$-player game where the local function of the $i$-th player is given by

$$f_i(x^i; x^{-i}) := \tfrac{1}{M} \sum_{m=1}^{M} f_{i,m}(x^i; x^{-i}), \tag{2}$$

for $i = 1, \ldots, n$ (with $d_1 = \cdots = d_n = d$). In this setting, each $f_{i,m}(x^i; x^{-i})$ takes the form $f_{i,m}(x^i; x^{-i}) = \frac{1}{2}\langle x^i, \mathbf{A}_{i,m} x^i \rangle + \sum_{1 \le j \le n, j \ne i} \langle x^i, \mathbf{B}_{i,j,m} x^j \rangle + \langle a_{i,m}, x^i \rangle$, where $\mathbf{A}_{i,m}, \mathbf{B}_{i,j,m} \in \mathbb{R}^{d \times d}$ and $a_{i,m} \in \mathbb{R}^d$ for $m = 1, \ldots, M$.

**Connection to game & control theory literature.** The above $n$-player game formulation has been often considered in game and control theory literature and has been used in recent works on distributed games (Nash equilibrium search) [117, 140, 80, 132, 60]. In connection with this literature, in Section 4.2, we demonstrate an experiment on a concrete robot control setup from [60].

We run PEARL-SGD with the theoretical step-size $\gamma = 1/(\ell\tau + 2(\tau-1)L_{\max}\sqrt{\kappa})$ from Theorems 3.3 and 3.4 with $\tau \in \{1, 2, 4, 5, 8, 20\}$. We set the cocoercivity parameter to $\ell = L^2/\mu$ following [33], where $L$ and $\mu$ are explicitly computed Lipschitz constant and strong monotonicity parameter of $\mathbb{F}$. Figure 2a displays the results from Deterministic PEARL-SGD, where we observe that all values of $\tau$ produce indistinguishable performance plots (which is predicted, as $\gamma$ scales down with $\tau$). Figure 2b shows results from the stochastic setting (we mini-batch from the finite sum (2)), where we repeat each experiment 5 times and plot the mean relative error with standard deviation (shaded region). It demonstrates that PEARL-SGD with larger synchronization interval $\tau$ provides a clear benefit of achieving smaller relative error $\frac{\|\mathbf{x}_k - \mathbf{x}_\star\|^2}{\|\mathbf{x}_0 - \mathbf{x}_\star\|^2}$ using the same number of communication rounds. These results verify our theoretical predictions from Section 3.

In Appendix E.1, we provide additional simulations regarding the case where the precise theoretical parameters $\mu, \ell, L_{\max}$ are not known, so that $\gamma$ has to be tuned empirically. It demonstrates that in practice, $(\tau, \gamma)$ can be effective tunable hyperparameters for gaining communication efficiency.

**Performance of PEARL-SGD for different $(\gamma, \tau)$ pairs.** Figure 3 displays the heatmap of relative errors (log-scale) after 100 communication rounds of Deterministic PEARL-SGD in the case of quadratic game with $n = 2$. White and yellow regions indicate divergence/poor performance; darker regions indicate lower relative errors.

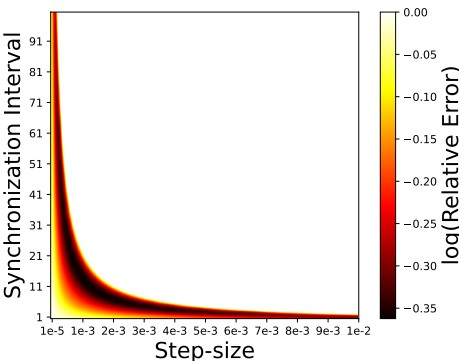

Figure 3: Heatmap of relative errors in logarithmic scale.

Figure 3 reveals a trend—for a fixed $\gamma$, PEARL-SGD's performance improves as $\tau$ increases up to certain threshold, after which it declines and finally diverges. Another key observation is that the dark region of the heatmap (signifying the best performance) takes the shape of a hyperbola. This is consistent with our Theorem 3.3, showing the relationship $\gamma_\tau \propto {^1/_\tau}$ where $\gamma_\tau$ is the optimal step-size choice given $\tau$ (providing fastest convergence).

### 4.2 Distributed mobile robot control

Here, we consider a distributed control problem of mobile robots from [60]. This is a multi-agent system where each robot has its own objective, depending on the positions $x^i \in \mathbb{R}^d$ (corresponding to action/strategy in our formulation of multiplayer game) of each $i$-th robot. Specifically, the objective function of the robot $i$ is:

$$f_i(\mathbf{x}) = \underbrace{\frac{a_i}{2}\|x^i - x^i_{\text{anc}}\|^2}_{:=J_{i1}(x^i)} + \underbrace{\frac{b_i}{2}\sum_{j=1}^N \|x^i - x^j - h_{ij}\|^2}_{:=J_{i2}(x^i;x^{-i})} \tag{3}$$

where $J_{i1}(x^i)$ represents the cost penalizing the distance of agent $i$ from the anchor point $x^i_{\text{anc}} \in \mathbb{R}^d$, and $J_{i2}(x^i; x^{-i})$ is the cost associated with the relative displacement between the robots' positions. The control problem finds an equilibrium of the $n$-player game, which is the concatenation of all robots' position vectors, ensuring that each robot stays close to $x^i_{\text{anc}}$ while maintaining designated displacement from other robots.

We implement PEARL-SGD with synchronization intervals $\tau \in \{1, 2, 4, 5, 8, 20\}$ and the theoretical step-size $\gamma = \frac{1}{\ell\tau + L_{\max}(\tau-1)\sqrt{\kappa}}$. Figure 2c shows that with larger values of $\tau$, PEARL-SGD achieves better accuracy (in terms of distance to $\mathbf{x}_\star$) within a given number of communication rounds. This highlights the potential benefit of using local update steps in solving real-world problems formulated as multiplayer games.

## 5  Conclusion

In this paper, we introduce Multiplayer Federated Learning (MpFL), an FL framework that models setups where clients, acting strategically in their own interests, collaborate through a central server to train models (actions) with the goal of reaching an equilibrium. We propose the PEARL-SGD algorithm for handling MpFL and establish its tight convergence guarantees under heterogeneous settings where each player has distinct objectives and data distributions. We show that PEARL-SGD achieves improved communication efficiency, mitigating the primary computational bottleneck in large-scale applications.

Our work offers several promising future research directions, including the incorporation of ideas such as Extragradient methods [68, 43], asynchronous updates [23, 127], gradient compression [4], gradient tracking [107], and algorithmic correction for client drifts [61, 100]. We anticipate that our initiation of the study of MpFL will stimulate further research along these and related directions.

## Acknowledgments

Sayantan Choudhury acknowledges support from the Chen Family Dissertation Fellowship. Nicolas Loizou acknowledges support from CISCO Research and the JHU Catalyst Award.

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

# Supplementary Material

We organize the appendix as follows: Section A provides an additional survey of related work. Section B provides a detailed explanation on how MpFL differs from prior FL frameworks. Section C presents the proofs of theoretical results. Section D provides the details of the experiments omitted from the main paper. Section E presents some additional experiments. Section F provides a detailed explanation and interpretation on the theoretical assumptions made in the paper.

## Contents

# A   Additional related work

**Game Theory & Equilibrium Computation.**   Multiplayer games, where multiple players each minimize their own cost function that is affected by the actions of the others, are a long-studied fundamental topic in mathematics and economics [106, 105, 120, 118, 70, 53, 90, 69, 88, 133]. More recently, there has been an increasing interest in the ML community in game-theoretic problems with motivating applications, including adversarial learning [40, 22], multi-agent reinforcement learning (MARL) [71, 79, 125], and language models [37, 57]. This incoming stream of applications has led to the development of novel analyses and insights regarding classical equilibrium-searching algorithms including gradient descent-ascent [24, 81, 136, 36, 147, 89, 72], extragradient [68, 18, 32, 74, 102, 99, 42, 43], optimistic gradient [109, 111, 112, 21, 39, 44] and consensus optimization/Hamiltonian gradient method [97, 7, 1, 89], and even the discovery of new accelerated algorithms for games [27, 142, 73, 16, 12, 143].

**Learning in games.**   Without local updates (the case $\tau = 1$), PEARL-SGD corresponds to the stochastic gradient play dynamics or the online gradient descent considered in the literature on learning in games [92, 82, 54], or more broadly, regularized Robbins-Monro processes or Follow-the-Generalized-Leader algorithms [96, 38]. Our setup considers the pure Nash equilibrium search in games with continuous and unconstrained action spaces, similarly as in [82]. The theoretical assumptions used in our analysis are similar to the ones that has appeared in this line of works; e.g., [82] uses strong monotonicity and cocoercivity of the joint gradient operator, while we use the weaker notion of quasi-strong monotonicity, which is similar to (but stronger than) the variational stability assumed in multiple works including [14, 95, 96], to name a few. Despite the close connections, our paper is distinguished from these works as we focus on communication efficiency in distributed optimization (federated learning) setup.

**Game theory for social client behavior in FL.**   Some prior work have also considered games with strategic clients participating in FL, focusing on designing mechanisms to prevent clients' social behaviors such as free-riding [104], coalitions [30] or dishonesty [31]. In these works, however, the meaning of the action $x_i$ is completely different from our setup of interest: it represents the size of the dataset which each client contributes to FL [11, 62, 104], strategy to deceive others and defend against those attacks [31], estimate of the trustworthiness of other clients [6] or willingness to participate in FL [17], all having restrictive meaning in a specific social context. On the other hand, in MpFL, $x_i$ are clients' *local models to be optimized through learning*, where the objectives $f_i$ follow *general game-theoretic structure*, and our primary focus is to design communication-efficient algorithms for finding an equilibrium.

**Heterogeneity and client drift.**   One fundamental challenge for theory of Local SGD (FedAvg) is heterogeneity, i.e., varying $f_i$'s due to differences in local data distributions [67, 78]. Under such setup, Local SGD is prone to client drift [146, 61] where local descent trajectories head toward distinct minima (of local objectives), and convergence theories require either additional assumptions [134, 144, 48, 78] or technical analyses [63, 65] to control this drift. Some papers, based on theoretical insights, introduced or analyzed correction mechanisms for Local SGD to mitigate client drift [61, 41, 101, 100, 55, 47]. Extension of such ideas to federated minimax optimization was explored in [145]. We note that the $n$-player game setup of MpFL is also fully heterogeneous as each player has distinct (possibly even conflicting) objective functions, and consequently, we have the analogous concept of *player drift*. We refer the readers interested in this topic to the discussion at the end of Section 3.2.

**FL frameworks with individual models.**   There are several distinct contexts for FL frameworks (other than personalized FL) where each client learns an individual model. In Vertical FL [137, 86, 87] scenarios, multiple organizations hold distinct features from the common set of samples and they collaborate to train their each local model. In Federated Transfer Learning [123, 85, 35], the participating organizations similarly keep and train local models, but their datasets have heterogeneity over both sample and feature spaces with limited overlaps. Federated Multi-Task Learning [124, 91, 98] extends FL to cases where each client solves different, but related tasks.

**Fictitious play.**   The Fictitious Play (FP) is a classical algorithm, originally proposed by [15] to solve minimax games where each player has a finite action space and plays mixed (randomized)

strategies. In FP, each player selects an action that minimizes their expected loss (best response), assuming that the other player plays the empirical (historical) strategy, which is a uniform random mixture of their previously played actions. The convergence of FP to a Nash equilibrium for minimax games was established in [113], but FP fails to converge for general $n$-player (with $n > 2$) or non-zero-sum games [121, 58], except for particular cases such as all players having identical objectives [103].

While it may appear that PEARL-SGD is conceptually similar to FP (as each player performing multiple local SGD steps can be interpreted as seeking a local approximate best response to others' strategies) the connection is opaque due to some fundamental differences. First, in PEARL-SGD, players make their updates based on only the most recent strategies of other players (not the entire history as in FP). Second, in PEARL-SGD, local SGD steps are not run until players converge to local optima—this results in player drift as we discuss at the end of Section 3.2, and is rather avoided by using step-sizes scaling down with the number of local steps. Third, in the FP setting players are assumed to have finite action spaces and mixed strategies (corresponding to points on a probability simplex), while the MpFL setting deals with continuous action spaces with pure (non-random) strategies. However, despite distinctions, as FP has been previously studied in the distributed $n$-player game setup [119], exploring the further connection between MpFL and FP could be an interesting direction.

**Federated bilevel optimization.** Bilevel optimization is a nested problem in which the outer optimization objective depends on the solution to an inner optimization problem [13, 3]. It can be viewed as a hierarchical game between a leader and a follower and generalizes minimax optimization. Recently, several works have studied bilevel formulations in federated learning (FL) settings [131, 56, 75, 110], with a growing focus on designing communication-efficient, single-loop algorithms for federated bilevel optimization [138, 139].

**Decoupled SGD.** The concurrent work [148] proposes and analyzes the *Decoupled SGD* algorithm whose update rule coincides with PEARL-SGD. While the exposition of this work emphasizes Decoupled SGDA — a version for two-player games, the multiplayer case is considered in their Appendix C. The paper defines $\mathbb{F}_{\mathbf{y}}(\mathbf{x}) = \left( \nabla f_1(x^1; y^{-1}), \ldots, \nabla f_n(x^n; y^{-n}) \right)$ for $\mathbf{y} \in \mathbb{R}^D$, and assuming that $\mathbb{F}_{\mathbf{y}}$ is $\bar{\mu}$-strongly monotone and $\|\mathbb{F}_{\mathbf{y}}(\mathbf{x}) - \mathbb{F}(\mathbf{x})\| \leq L_c \|\mathbf{x} - \mathbf{y}\|, \forall \mathbf{x}, \mathbf{y} \in \mathbb{R}^D$, shows that Decoupled SGD can provide communication acceleration in the *weakly-coupled* regime where $\kappa_c = L_c / \bar{\mu}$ is small (even in the deterministic regime). Despite the algorithmic similarity between PEARL-SGD and Decoupled SGD, our work differs from [148] not only in terms of how technical results are derived, but also in terms of the emphasis. In particular, unlike [148], we consider PEARL-SGD as a component of the broader framework of MpFL, which we view as our most significant conceptual contribution.

# B  Detailed distinction of MpFL from prior FL setups

**Classical FL algorithms are incompatible with MpFL.**   We first clarify that classical FL algorithms including Local SGD [94], FedProx [78], SCAFFOLD [61] or Scaffnew [100] are not suitable for the MpFL setting. There the problem is formulated as

$$\underset{x \in \mathbb{R}^d}{\text{minimize}} \quad f(x) = \frac{1}{n} \sum_{i=1}^{n} f_i(x). \tag{FL}$$

On the other hand, recall that MpFL is formulated as

$$\underset{\mathbf{x}_\star = (x_\star^1, \dots, x_\star^n) \in \mathbb{R}^D}{\text{find}} \quad f_i(x_\star^i; x_\star^{-i}) \leq f_i(x^i; x_\star^{-i}), \quad \forall x^i \in \mathbb{R}^{d_i}, \quad \forall i \in [n]. \tag{MpFL}$$

One obvious distinction is that (FL) seeks a single $x_\star \in \mathbb{R}^d$ minimizing the finite sum, while in (MpFL) each player finds distinct $x^1, \dots, x^n$ satisfying the equilibrium condition. The player $i$ **does not** update $x^j$ for $j \neq i$ in MpFL, so unlike in classical FL, each player only contributes to partial coordinates of the desired solution $\mathbf{x}_\star = (x_\star^1, \dots, x_\star^n)$. Therefore, we cannot apply Local SGD or its variants to the MpFL setup.

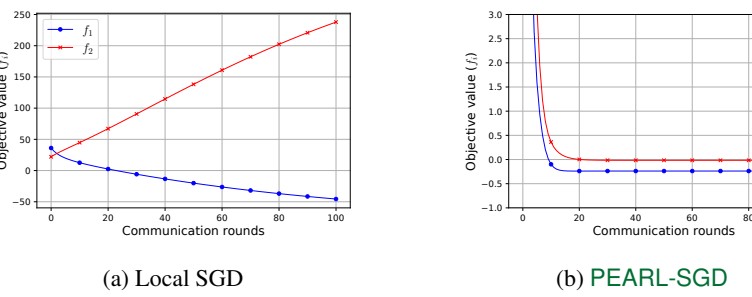

(a) Local SGD

(b) PEARL-SGD

Figure 4: Plots of objective values $f_1$, $f_2$ in (4) from running **(left)** Local SGD on the joint variable $(u, v)$ and **(right)** PEARL-SGD.

Additionally, it is generally not possible to reach an equilibrium by performing SGD on the sum of objectives (as in Local SGD). Consider the following simple example with $n = 2$ clients:

$$f_1(u; v) = \frac{1}{2} u^\mathsf{T} (\mathbf{A}u - a - \mathbf{B}^\mathsf{T} v) - \frac{1}{20} \|v\|^2, \quad f_2(v; u) = \frac{1}{4} \|v\|^2 + \frac{1}{2} v^\mathsf{T} (\mathbf{B}u - b) - \frac{1}{20} \|u\|^2 \tag{4}$$

where $u, v \in \mathbb{R}^d$, $\mathbf{A} \succ 0$ and $a, b \in \mathbb{R}^d$ (we use $(u, v)$ instead of $(x^1, x^2)$ for clearer notation). Running Local SGD with respect to the joint variable $(u, v)$ on the sum $\frac{1}{2}(f_1(u, v) + f_2(u, v))$ results in divergence of one of the objective values (Fig. 4a) while PEARL-SGD converges to the equilibrium and the objective values stabilize (Fig. 4b). However, although we included this example for illustration, note that here Local SGD is not even conforming to the rules of MpFL as both clients are updating $(u, v)$ simultaneously.

**Federated minimax optimization (FMO) algorithms are incompatible with MpFL.**   In FMO, the problem is

$$\underset{x \in \mathbb{R}^{d_x}}{\text{minimize}} \underset{y \in \mathbb{R}^{d_y}}{\text{maximize}} \quad \mathcal{L}(x, y) = \frac{1}{n} \sum_{i=1}^{n} \mathcal{L}_i(x, y). \tag{FMO}$$

In algorithms designed for (FMO) including Local SGDA or Local SEG, each client $i$ locally updates *both* the minimization and maximization variables $(x^i, y^i)$, and *all clients work collaboratively* toward finding the minimax solution (saddle point) of $\mathcal{L}(x, y)$, which is a global objective. On the other hand, in (MpFL), each player $i$ locally updates *only* their own action $x^i$, *for their own individual interest* of reducing $f_i(\cdot; x^{-i})$. Even though both (FMO) and (MpFL) aim to reach an equilibrium, they are completely different processes from a conceptual level. For example, for the $n$-player game setup in Section 4.1 with $n = 5$, it is not possible to apply Local SGDA or Local SEG due to the conceptual mismatch.

# C Omitted proofs for Per-Player Local SGD (PEARL-SGD)

## C.1 Key ideas and proof outline

We first provide an outline for the proof of Theorem 3.4. The key components of the proof are as follows: **(i)** we show that a round of local SGD in PEARL-SGD behaves like a large single descent step with respect to the joint gradient operator $\mathbb{F}$ except for *local error* terms caused by running multiple SGD steps locally (Lemma C.1), and **(ii)** we bound these local error terms (Lemma C.2).

> **Lemma C.1.** Assume *(SM)*, and let $L_{\max} = \max\{L_1, \ldots, L_n\}$. Let $0 \le p \le R - 1$ be a fixed round index in PEARL-SGD and suppose $\gamma_k \equiv \gamma > 0$ for $k = \tau p, \ldots, \tau(p+1) - 1$. Then for arbitrary $\alpha > 0$, we have
>
> $$\mathbb{E}\left[\left\|\mathbf{x}_{\tau(p+1)} - \mathbf{x}_\star\right\|^2 \Big| \mathbf{x}_{\tau p}\right] \le (1 + (\tau - 1)\alpha\gamma) \left\|\mathbf{x}_{\tau p} - \mathbf{x}_\star\right\|^2 - 2\gamma\tau \left\langle \mathbb{F}(\mathbf{x}_{\tau p}), \mathbf{x}_{\tau p} - \mathbf{x}_\star \right\rangle$$
> $$+ \frac{\gamma L_{\max}^2}{\alpha} \sum_{j=\tau p+1}^{\tau p + \tau - 1} \mathbb{E}\left[\left\|\mathbf{x}_{\tau p} - \mathbf{x}_j\right\|^2 \Big| \mathbf{x}_{\tau p}\right] + \mathbb{E}\left[\left\|\mathbf{x}_{\tau p} - \mathbf{x}_{\tau(p+1)}\right\|^2 \Big| \mathbf{x}_{\tau p}\right].$$

**Local error bound.** The right hand side of the bound in Lemma C.1 involves the quantities

$$\mathbb{E}\left[\left\|\mathbf{x}_{\tau p} - \mathbf{x}_j\right\|^2 \Big| \mathbf{x}_{\tau p}\right] = \sum_{i=1}^n \mathbb{E}\left[\left\|x_{\tau p}^i - x_j^i\right\|^2 \Big| \mathbf{x}_{\tau p}\right] \tag{5}$$

for $j = \tau p + 1, \ldots, \tau(p+1)$. We further bound (5) using the following result.

> **Lemma C.2.** Suppose Assumptions *(CVX)*, *(SM)* and *(BV)* hold. For a fixed $i \in [n]$ and a fixed communication round $p$ in PEARL-SGD, suppose $\gamma_k \equiv \gamma$ for $k = \tau p, \ldots, \tau(p+1) - 1$, where $0 < \gamma \le \frac{1}{L_i} \min\left\{1, \frac{1}{\tau-1}\right\}$. Then for $t = 0, \ldots, \tau$,
>
> $$\mathbb{E}\left[\left\|x_{\tau p}^i - x_{\tau p + t}^i\right\|^2 \Big| \mathbf{x}_{\tau p}\right] \le \gamma^2 t^2 \left\|\nabla f(x_{\tau p}^i; x_{\tau p}^{-i})\right\|^2 + \gamma^2 t \left(1 + 2(t-1)(t+1)\gamma L_i\right)\sigma_i^2.$$

Here we sketch the proof of Lemma C.2 and clarify the role of Assumption *(CVX)*. By assuming that each $f_i(\cdot; x_{\tau p}^{-i})$ is convex and $L_i$-smooth, we can prove Lemma C.3, showing that the expectation of squared gradient norm is "almost" nonincreasing along the local SGD steps, except for some additional term due to stochasticity. Then, we rewrite each summand in (5) as

$$\mathbb{E}\left[\left\|x_{\tau p}^i - x_k^i\right\|^2 \Big| \mathbf{x}_{\tau p}\right] = \mathbb{E}\left[\gamma^2 \left\|\sum_{j=\tau p}^{k-1} g_j^i\right\|^2 \Big| \mathbf{x}_{\tau p}\right] = \mathbb{E}\left[\gamma^2 \left\|\sum_{j=\tau p}^{k-1} \nabla f_{i,\xi_j^i}(x_j^i; x_{\tau p}^{-i})\right\|^2 \Big| \mathbf{x}_{\tau p}\right] \tag{6}$$

and use Lemma C.3 to bound (6).

> **Lemma C.3.** Under the assumptions of Lemma C.2, for $j = \tau p + 1, \ldots, \tau(p+1)$,
>
> $$\mathbb{E}\left[\left\|\nabla f_i(x_j^i; x_{\tau p}^{-i})\right\|^2 \Big| \mathbf{x}_{\tau p}\right] \le \left\|\nabla f_i(x_{\tau p}^i; x_{\tau p}^{-i})\right\|^2 + 2(j - \tau p)\gamma L_i \sigma_i^2.$$

**Remark.** Given (6), it is tempting to apply Jensen's inequality to the rightmost quantity and then apply Lemma C.3. However, this results in a bound that is looser than our Lemma C.2. We need more careful arguments regarding the expectations, which we detail throughout Appendix C.

*Proof outline for Theorem 3.4.* We combine Lemmas C.1 and C.2, and then apply *(SCO)* to eliminate the $\|\mathbf{F}(\mathbf{x}_{\tau p})\|^2$ terms to obtain

$$\mathbb{E}\left[\left\|\mathbf{x}_{\tau(p+1)} - \mathbf{x}_\star\right\|^2 \Big| \mathbf{x}_{\tau p}\right] \le (1 + (\tau - 1)\alpha\gamma) \|\mathbf{x}_{\tau p} - \mathbf{x}_\star\|^2 + \text{(terms proportional to } \sigma^2)$$
$$- \underbrace{\left(2\gamma\tau - \gamma^2\tau^2\ell - \frac{\gamma^3 L_{\max}^2 \tau^2(\tau - 1)\ell}{3\alpha}\right)}_{:=C} \langle \mathbf{F}(\mathbf{x}_{\tau p}), \mathbf{x}_{\tau p} - \mathbf{x}_\star \rangle \quad (7)$$

Provided that $C \ge 0$, we can upper bound the second line of (7) by $-C\mu \|\mathbf{x}_{\tau p} - \mathbf{x}_\star\|^2$ using *(QSM)*. Then we choose $\alpha = \gamma\tau L_{\max}\sqrt{\frac{\ell\mu}{3}}$ which minimizes the resulting coefficient of $\|\mathbf{x}_{\tau p} - \mathbf{x}_\star\|^2$, and rewrite it in the form $1 - \gamma\tau\mu\zeta$. Finally, take expectation over $\mathbf{x}_{\tau p}$ in (7) and unroll the recursion. $\square$

The proofs of Lemmas C.1, C.2 and C.3 and the detailed full proof of Theorem 3.4 are presented through the following subsections.

## C.2 Proof of Lemma C.1

Note that for $k = \tau p + 1, \ldots, \tau(p + 1)$ (iterations between $p$-th and $(p + 1)$-th communications), we have

$$\|\mathbf{x}_k - \mathbf{x}_\star\|^2 = \sum_{i=1}^{n} \left\|x_k^i - x_\star^i\right\|^2$$
$$= \sum_{i=1}^{n} \left\|x_{\tau p}^i - x_\star^i - \left(x_{\tau p}^i - x_k^i\right)\right\|^2$$
$$= \sum_{i=1}^{n} \left[\left\|x_{\tau p}^i - x_\star^i\right\|^2 - 2\left\langle x_{\tau p}^i - x_\star^i, x_{\tau p}^i - x_k^i\right\rangle + \left\|x_{\tau p}^i - x_k^i\right\|^2\right]$$
$$= \|\mathbf{x}_{\tau p} - \mathbf{x}_\star\|^2 - 2\gamma \sum_{i=1}^{n} \sum_{j=\tau p}^{k-1} \left\langle x_{\tau p}^i - x_\star^i, g_j^i\right\rangle + \sum_{i=1}^{n} \left\|x_{\tau p}^i - x_k^i\right\|^2, \quad (8)$$

where for the last equality, we use

$$g_j^i = \nabla f_{i,\xi_j^i}(x_j^i; x_{\tau p}^{-i}), \quad j = \tau p, \ldots, k - 1, \quad i = 1, \ldots, n$$

and

$$x_{j+1}^i = x_j^i - \gamma g_j^i, \quad j = \tau p, \ldots, k - 1, \quad i = 1, \ldots, n$$

to rewrite $x_{\tau p}^i - x_k^i = \gamma \sum_{j=\tau p}^{k-1} g_j^i$. Note that we have

$$\mathbb{E}\left[-\left\langle x_{\tau p}^i - x_\star^i, g_{\tau p}^i\right\rangle \Big| \mathbf{x}_{\tau p}\right] = -\left\langle x_{\tau p}^i - x_\star^i, \nabla f_i(x_{\tau p}^i; x_{\tau p}^{-i})\right\rangle,$$

while for the other indices $j = \tau p + 1, \ldots, k - 1$, we have the upper bound

$$\mathbb{E}\left[-\left\langle x_{\tau p}^i - x_\star^i, g_j^i\right\rangle \Big| x_j^i\right]$$
$$= -\left\langle x_{\tau p}^i - x_\star^i, \nabla f_i(x_j^i; x_{\tau p}^{-i})\right\rangle$$
$$= -\left\langle x_{\tau p}^i - x_\star^i, \nabla f_i(x_{\tau p}^i; x_{\tau p}^{-i})\right\rangle + \left\langle x_{\tau p}^i - x_\star^i, \nabla f_i(x_{\tau p}^i; x_{\tau p}^{-i}) - \nabla f_i(x_j^i; x_{\tau p}^{-i})\right\rangle$$
$$\le -\left\langle x_{\tau p}^i - x_\star^i, \nabla f_i(x_{\tau p}^i; x_{\tau p}^{-i})\right\rangle + \frac{\alpha}{2}\left\|x_{\tau p}^i - x_\star^i\right\|^2 + \frac{1}{2\alpha}\left\|\nabla f_i(x_{\tau p}^i; x_{\tau p}^{-i}) - \nabla f_i(x_j^i; x_{\tau p}^{-i})\right\|^2$$
$$\le -\left\langle x_{\tau p}^i - x_\star^i, \nabla f_i(x_{\tau p}^i; x_{\tau p}^{-i})\right\rangle + \frac{\alpha}{2}\left\|x_{\tau p}^i - x_\star^i\right\|^2 + \frac{L_i^2}{2\alpha}\left\|x_{\tau p}^i - x_j^i\right\|^2$$

where in the fourth line, we use Young's inequality with an arbitrary $\alpha > 0$ that we determine later. Take expectations of the both sides in (8) (conditioned on $\mathbf{x}_{\tau p}$), and apply the above bound with the

tower rule to obtain

$$
\mathbb{E}\left[\left\|\mathbf{x}_k - \mathbf{x}_\star\right\|^2 \,\Big|\, \mathbf{x}_{\tau p}\right]
$$

$$
\leq \left\|\mathbf{x}_{\tau p} - \mathbf{x}_\star\right\|^2 - 2\gamma \sum_{i=1}^{n}\sum_{j=\tau p}^{k-1} \left\langle x_{\tau p}^i - x_\star^i, \nabla f_i(x_{\tau p}^i; x_{\tau p}^{-i})\right\rangle + 2\gamma \sum_{i=1}^{n}\sum_{j=\tau p+1}^{k-1} \frac{\alpha}{2}\left\|x_{\tau p}^i - x_\star^i\right\|^2 \tag{9}
$$

$$
+ 2\gamma \sum_{i=1}^{n}\sum_{j=\tau p+1}^{k-1} \mathbb{E}\left[\frac{L_i^2}{2\alpha}\left\|x_{\tau p}^i - x_j^i\right\|^2 \,\Big|\, \mathbf{x}_{\tau p}\right] + \sum_{i=1}^{n}\mathbb{E}\left[\left\|x_{\tau p}^i - x_k^i\right\|^2 \,\Big|\, \mathbf{x}_{\tau p}\right].
$$

Now we apply the identities

$$
\sum_{i=1}^{n}\left\langle x_{\tau p}^i - x_\star^i, \nabla f_i(x_{\tau p}^i; x_{\tau p}^{-i})\right\rangle = \left\langle \mathbf{x}_{\tau p} - \mathbf{x}_\star, \mathbb{F}(\mathbf{x}_{\tau p})\right\rangle, \quad \sum_{i=1}^{n}\left\|x_{\tau p}^i - x_\star^i\right\|^2 = \left\|\mathbf{x}_{\tau p} - \mathbf{x}_\star\right\|^2
$$

$$
\sum_{i=1}^{n}\mathbb{E}\left[\left\|x_{\tau p}^i - x_k^i\right\|^2 \,\Big|\, \mathbf{x}_{\tau p}\right] = \mathbb{E}\left[\left\|\mathbf{x}_{\tau p} - \mathbf{x}_k\right\|^2 \,\Big|\, \mathbf{x}_{\tau p}\right]
$$

and the inequality

$$
\sum_{i=1}^{n}\sum_{j=\tau p+1}^{k-1} \mathbb{E}\left[\frac{L_i^2}{2\alpha}\left\|x_{\tau p}^i - x_j^i\right\|^2 \,\Big|\, \mathbf{x}_{\tau p}\right] \leq \frac{L_{\max}^2}{2\alpha}\sum_{j=\tau p+1}^{k-1}\sum_{i=1}^{n}\mathbb{E}\left[\left\|x_{\tau p}^i - x_j^i\right\|^2 \,\Big|\, \mathbf{x}_{\tau p}\right]
$$

$$
= \frac{L_{\max}^2}{2\alpha}\sum_{j=\tau p+1}^{k-1}\mathbb{E}\left[\left\|\mathbf{x}_{\tau p} - \mathbf{x}_j\right\|^2 \,\Big|\, \mathbf{x}_{\tau p}\right]
$$

to (9) and plug in $k = \tau(p+1)$, which gives the desired result.

### C.3   General properties and bounds for SGD

In this section, we present some general properties of stochastic gradient descent (SGD) for an $L$-smooth, convex function $f \colon \mathbb{R}^m \to \mathbb{R}$. Suppose that we have a stochastic oracle $\nabla f_\xi(\cdot)$ for the gradient operator $\nabla f(\cdot)$, satisfying

$$
\mathbb{E}_\xi[\nabla f_\xi(x)] = \nabla f(x), \quad \mathbb{E}_\xi\left[\left\|\nabla f_\xi(x) - \nabla f(x)\right\|^2\right] \leq \rho^2, \quad \forall x \in \mathbb{R}^m \tag{10}
$$

where $\mathbb{E}_\xi$ denotes the expectation with respect to randomness in $\xi$. This setup and the subsequent results are the abstractions of intermediate results that we need for the proofs of Lemma C.3 and Lemma C.2. Specifically, we will later use the results of this section with

$$
f(\cdot) = f_i(\cdot; x_{\tau p}^{-i}), \qquad \rho^2 = \sigma_i^2,
$$

for each $i = 1, \ldots, n$. We make this abstraction to simplify notations and to more effectively convey the key intuitions underlying the analyses.

---

**Lemma C.4.** Let $f \colon \mathbb{R}^m \to \mathbb{R}$ be convex and $L$-smooth. Suppose that a stochastic gradient oracle $\nabla f_\xi(\cdot)$ satisfies (10). Let $y = x - \gamma \nabla f_\xi(x)$, where $0 < \gamma \leq \frac{2}{L}$. Then we have

$$
\mathbb{E}_\xi\left[\left\|\nabla f(y)\right\|^2\right] \leq \left\|\nabla f(x)\right\|^2 + 2\gamma L \rho^2.
$$

---

*Proof.* It is well-known that if $f$ is convex and $L$-smooth, then $\nabla f$ is $\frac{1}{L}$-cocoercive, i.e., for any $x, y \in \mathbb{R}^m$,

$$
\left\langle x - y, \nabla f(x) - \nabla f(y)\right\rangle \geq \frac{1}{L}\left\|\nabla f(x) - \nabla f(y)\right\|^2.
$$

By cocoercivity and the step-size condition $\gamma \leq \frac{2}{L}$, we have

$$\frac{\gamma}{2} \|\nabla f(x) - \nabla f(y)\|^2$$

$$\leq \frac{1}{L} \|\nabla f(x) - \nabla f(y)\|^2$$

$$\leq \langle x - y, \nabla f(x) - \nabla f(y) \rangle$$

$$= \langle \gamma \nabla f_\xi(x), \nabla f(x) - \nabla f(y) \rangle$$

$$= \gamma \left( \langle \nabla f_\xi(x), \nabla f(x) \rangle - \langle \nabla f(x), \nabla f(y) \rangle + \langle \nabla f(x) - \nabla f_\xi(x), \nabla f(y) \rangle \right).$$

Taking expectation of the both sides, we obtain

$$\mathbb{E}_\xi \left[ \frac{\gamma}{2} \|\nabla f(x) - \nabla f(y)\|^2 \right]$$

$$\leq \mathbb{E}_\xi \left[ \gamma \langle \nabla f_\xi(x), \nabla f(x) \rangle - \gamma \langle \nabla f(x), \nabla f(y) \rangle + \gamma \langle \nabla f(x) - \nabla f_\xi(x), \nabla f(y) \rangle \right]$$

$$= \gamma \|\nabla f(x)\|^2 - \gamma \mathbb{E}_\xi \left[ \langle \nabla f(x), \nabla f(y) \rangle \right] + \gamma \mathbb{E}_\xi \left[ \langle \nabla f(x) - \nabla f_\xi(x), \nabla f(y) \rangle \right].$$

Cancelling out the terms and dividing both sides by $\frac{\gamma}{2}$, we then have

$$\mathbb{E}_\xi \left[ \|\nabla f(y)\|^2 \right] \leq \|\nabla f(x)\|^2 + 2\mathbb{E}_\xi \left[ \langle \nabla f(x) - \nabla f_\xi(x), \nabla f(y) \rangle \right]. \tag{11}$$

Now observe that

$$\mathbb{E}_\xi \left[ \langle \nabla f(x) - \nabla f_\xi(x), \nabla f(y) \rangle \right] = \mathbb{E}_\xi \left[ \langle \nabla f(x) - \nabla f_\xi(x), \nabla f(y) - \nabla f(x - \gamma \nabla f(x)) \rangle \right]$$

because $\nabla f(x - \gamma \nabla f(x))$ is a non-random quantity and $\mathbb{E}_\xi [\nabla f(x) - \nabla f_\xi(x)] = 0$. Then

$$\mathbb{E}_\xi \left[ \langle \nabla f(x) - \nabla f_\xi(x), \nabla f(y) - \nabla f(x - \gamma \nabla f(x)) \rangle \right]$$

$$= \mathbb{E}_\xi \left[ \langle \nabla f(x) - \nabla f_\xi(x), \nabla f(x - \gamma \nabla f_\xi(x)) - \nabla f(x - \gamma \nabla f(x)) \rangle \right]$$

$$\leq \mathbb{E}_\xi \left[ \|\nabla f(x) - \nabla f_\xi(x)\| \, \|\nabla f(x - \gamma \nabla f_\xi(x)) - \nabla f(x - \gamma \nabla f(x))\| \right]$$

$$\leq \mathbb{E}_\xi \left[ \|\nabla f(x) - \nabla f_\xi(x)\| \, L \, \|(x - \gamma \nabla f_\xi(x)) - (x - \gamma \nabla f(x))\| \right]$$

$$= \gamma L \mathbb{E}_\xi \left[ \|\nabla f(x) - \nabla f_\xi(x)\|^2 \right]$$

$$= \gamma L \rho^2,$$

and plugging this into (11) completes the proof.

$\square$

**Lemma C.5.** Let $f : \mathbb{R}^m \to \mathbb{R}$ be convex and $L$-smooth and let the stochastic gradient oracle $\nabla f_\xi(\cdot)$ satisfy (10). Let $x_0 \in \mathbb{R}^m$ be any initial point, $0 < \gamma \leq \frac{2}{L}$, and $x_1, \ldots, x_t$ be a sequence generated by the stochastic gradient descent algorithm

$$x_{s+1} = x_s - \gamma \nabla f_{\xi_s}(x_s)$$

for $s = 0, \ldots, t - 1$. Then we have

$$\mathbb{E} \left[ \|\nabla f(x_s)\|^2 \right] \leq \|\nabla f(x_0)\|^2 + 2s\gamma L \rho^2$$

for $s = 0, \ldots, t - 1$.

*Proof.* Apply Lemma C.4 recursively and use the tower rule (law of total expectation).

$\square$

**Lemma C.6.** Let $f : \mathbb{R}^m \to \mathbb{R}$ be $L$-smooth and let $x_0, \ldots, x_t$ be a sequence generated by stochastic gradient descent

$$x_{s+1} = x_s - \gamma \nabla f_{\xi_s}(x_s)$$

where the stochastic gradient oracle satisfies (10). Let $\hat{x}_0, \ldots, \hat{x}_t$ be generated via *deterministic* gradient descent

$$\hat{x}_{s+1} = \hat{x}_s - \gamma \nabla f(\hat{x}_s)$$

where $\hat{x}_0 = x_0$. Then, provided that $0 < \gamma \leq \frac{1}{L(t-1)}$, we have

$$\|x_t - \hat{x}_t\| \leq 3\gamma \sum_{s=0}^{t-1} \|\nabla f_{\xi_s}(x_s) - \nabla f(x_s)\| .$$

**Remark.** This result only assumes $L$-smoothness of $f$ (which is $L$-Lipschitz continuity of $\nabla f$) and does not require convexity.

*Proof.* When $t = 1$, we have $\|x_t - \hat{x}_t\| = \gamma \|\nabla f_{\xi_0}(x_0) - \nabla f(x_0)\|$ as $x_0 = \hat{x}_0$.

Now assume $t > 1$. Observe that

$$x_t - \hat{x}_t = (x_{t-1} - \hat{x}_{t-1}) - \gamma \left( \nabla f_{\xi_{t-1}}(x_{t-1}) - \nabla f(\hat{x}_{t-1}) \right)$$
$$= (x_{t-1} - \hat{x}_{t-1}) - \gamma \left( \nabla f_{\xi_{t-1}}(x_{t-1}) - \nabla f(x_{t-1}) \right) - \gamma \left( \nabla f(x_{t-1}) - \nabla f(\hat{x}_{t-1}) \right)$$

and therefore,

$$\|x_t - \hat{x}_t\| \leq \|x_{t-1} - \hat{x}_{t-1}\| + \gamma \left\| \nabla f_{\xi_{t-1}}(x_{t-1}) - \nabla f(x_{t-1}) \right\| + \gamma \left\| \nabla f(x_{t-1}) - \nabla f(\hat{x}_{t-1}) \right\|$$
$$\leq (1 + \gamma L) \|x_{t-1} - \hat{x}_{t-1}\| + \gamma \left\| \nabla f_{\xi_{t-1}}(x_{t-1}) - \nabla f(x_{t-1}) \right\|$$

where the last inequality uses the $L$-smoothness assumption. Now unrolling the recursion and using the fact $\|x_0 - \hat{x}_0\| = 0$ we obtain

$$\|x_t - \hat{x}_t\| \leq \sum_{s=0}^{t-1} \gamma(1 + \gamma L)^{t-s-1} \|\nabla f_{\xi_s}(x_s) - \nabla f(x_s)\|$$

$$\leq \gamma \left( 1 + \frac{1}{t-1} \right)^{t-1} \sum_{s=0}^{t-1} \|\nabla f_{\xi_s}(x_s) - \nabla f(x_s)\|$$

$$\leq 3\gamma \sum_{s=0}^{t-1} \|\nabla f_{\xi_s}(x_s) - \nabla f(x_s)\| .$$

$\square$

**Lemma C.7.** Under the assumptions of Lemma C.6, we have

$$\mathbb{E}\left[ \langle \nabla f_{\xi_0}(x_0) - \nabla f(x_0), \nabla f(x_t) \rangle \right] \leq 3t\gamma L\rho^2 .$$

*Proof.* Observe that because $\hat{x}_t$ as defined in Lemma C.6 is a non-random quantity and
$$\mathbb{E}\left[ \nabla f_{\xi_0}(x_0) - \nabla f(x_0) \right] = 0,$$
we have
$$\mathbb{E}\left[ \langle \nabla f_{\xi_0}(x_0) - \nabla f(x_0), \nabla f(\hat{x}_t) \rangle \right] = 0.$$
Using this, we can proceed as in the following to obtain the desired bound:

$$\mathbb{E}\left[ \langle \nabla f_{\xi_0}(x_0) - \nabla f(x_0), \nabla f(x_t) \rangle \right]$$
$$= \mathbb{E}\left[ \langle \nabla f_{\xi_0}(x_0) - \nabla f(x_0), \nabla f(x_t) - \nabla f(\hat{x}_t) \rangle \right]$$
$$\leq \mathbb{E}\left[ \|\nabla f_{\xi_0}(x_0) - \nabla f(x_0)\| \|\nabla f(x_t) - \nabla f(\hat{x}_t)\| \right]$$
$$\leq \mathbb{E}\left[ \|\nabla f_{\xi_0}(x_0) - \nabla f(x_0)\| L \|x_t - \hat{x}_t\| \right]$$
$$\leq 3\gamma L \, \mathbb{E}\left[ \|\nabla f_{\xi_0}(x_0) - \nabla f(x_0)\| \sum_{s=0}^{t-1} \|\nabla f_{\xi_s}(x_s) - \nabla f(x_s)\| \right]$$
$$\leq 3\gamma L \, \mathbb{E}\left[ \sum_{s=0}^{t-1} \left( \frac{\|\nabla f_{\xi_0}(x_0) - \nabla f(x_0)\|^2}{2} + \frac{\|\nabla f_{\xi_s}(x_s) - \nabla f(x_s)\|^2}{2} \right) \right]$$
$$\leq 3t\gamma L\rho^2 .$$

$\square$

**Lemma C.8.** Under the assumptions of Lemma C.6, we have

$$\mathbb{E}\left[\|x_0 - x_t\|^2\right] \le \gamma^2 \mathbb{E}\left[\left\|\sum_{s=0}^{t-1} \nabla f(x_s)\right\|^2\right] + \gamma^2 t\rho^2 + (t-1)t(t+1)\gamma^3 L\rho^2.$$

*Proof.* In the case $t = 1$, we have

$$\mathbb{E}\left[\|x_0 - x_1\|^2\right] = \gamma^2 \mathbb{E}_{\xi_0}\left[\|\nabla f_{\xi_0}(x_0)\|^2\right] \le \gamma^2 \rho^2 + \gamma^2 \|\nabla f(x_0)\|^2,$$

which is the desired statement. Now we use induction on $t$. Suppose that the result holds for any initial point and $t$ steps of SGD. Consider a sequence $x_0, \dots, x_{t+1}$ generated via SGD with initial point $x_0$ and step-size $\gamma > 0$. Observe that

$$\mathbb{E}\left[\|x_0 - x_{t+1}\|^2\right]$$

$$= \gamma^2 \mathbb{E}\left[\left\|\sum_{s=0}^{t} \nabla f_{\xi_s}(x_s)\right\|^2\right]$$

$$= \gamma^2 \mathbb{E}\left[\left\|\sum_{s=0}^{t-1} \nabla f_{\xi_s}(x_s)\right\|^2 + \mathbb{E}_{\xi_t}\left[2\left\langle \nabla f_{\xi_t}(x_t), \sum_{s=0}^{t-1} \nabla f_{\xi_s}(x_s)\right\rangle + \|\nabla f_{\xi_t}(x_t)\|^2 \,\bigg|\, x_t\right]\right]$$

$$\le \mathbb{E}\left[\|x_0 - x_t\|^2\right] + \gamma^2 \mathbb{E}\left[2\left\langle \nabla f(x_t), \sum_{s=0}^{t-1} \nabla f_{\xi_s}(x_s)\right\rangle + \|\nabla f(x_t)\|^2 + \rho^2\right] \quad (12)$$

where the third line uses the tower rule. Now observe that for $s = 0, \dots, t-1$,

$$\mathbb{E}\left[\langle \nabla f(x_t), \nabla f_{\xi_s}(x_s)\rangle\right] = \mathbb{E}\left[\langle \nabla f(x_t), \nabla f(x_s)\rangle\right] + \mathbb{E}\left[\langle \nabla f(x_t), \nabla f_{\xi_s}(x_s) - \nabla f(x_s)\rangle\right]$$

$$= \mathbb{E}\left[\langle \nabla f(x_t), \nabla f(x_s)\rangle\right] + \mathbb{E}\left[\mathbb{E}\left[\langle \nabla f_{\xi_s}(x_s) - \nabla f(x_s), \nabla f(x_t)\rangle \,|\, x_s\right]\right]$$

$$\le \mathbb{E}\left[\langle \nabla f(x_t), \nabla f(x_s)\rangle\right] + 3(t-s)\gamma L\rho^2$$

where the last inequality uses Lemma C.7 (with $x_s$ regarded as initial point of the stochastic gradient descent). Now we apply this inequality and the induction hypothesis to (12):

$$\mathbb{E}\left[\|x_0 - x_{t+1}\|^2\right]$$

$$\le \gamma^2 \mathbb{E}\left[\left\|\sum_{s=0}^{t-1} \nabla f(x_s)\right\|^2 + t\rho^2 + (t-1)t(t+1)\gamma L\rho^2\right.$$

$$\left. + \sum_{s=0}^{t-1}\left(2\langle \nabla f(x_t), \nabla f(x_s)\rangle + 6(t-s)\gamma L\rho^2\right) + \|\nabla f(x_t)\|^2 + \rho^2\right]$$

$$= \gamma^2\left(t\rho^2 + (t-1)t(t+1)\gamma L\rho^2 + 3t(t+1)\gamma L\rho^2 + \rho^2\right)$$

$$+ \gamma^2 \mathbb{E}\left[\left\|\sum_{s=0}^{t-1} \nabla f(x_s)\right\|^2 + 2\left\langle \sum_{s=0}^{t-1} \nabla f(x_s), \nabla f(x_t)\right\rangle + \|\nabla f(x_t)\|^2\right]$$

$$= \gamma^2(t+1)\rho^2 + t(t+1)(t+2)\gamma^3 L\rho^2 + \gamma^2 \mathbb{E}\left[\left\|\sum_{s=0}^{t} \nabla f(x_s)\right\|^2\right]$$

where for the first equality we use $\sum_{s=0}^{t-1} 6(t-s) = 3t(t+1)$. This completes the induction.

$\square$

**Lemma C.9.** Let $f \colon \mathbb{R}^m \to \mathbb{R}$ be convex and $L$-smooth, and let $x_0 \in \mathbb{R}^m$ be any initial point. Let $x_1, \ldots, x_t$ be generated by stochastic gradient descent

$$x_{s+1} = x_s - \gamma \nabla f_{\xi_s}(x_s)$$

with $0 < \gamma \le \frac{1}{L} \min\left\{1, \frac{1}{t-1}\right\}$. Then

$$\mathbb{E}\left[\|x_0 - x_t\|^2\right] \le \gamma^2 t^2 \|\nabla f(x_0)\|^2 + \gamma^2 t(1 + 2(t-1)(t+1)\gamma L)\rho^2.$$

*Proof.* Lemma C.8 gives

$$\mathbb{E}\left[\|x_0 - x_t\|^2\right] \le \gamma^2 \mathbb{E}\left[\left\|\sum_{s=0}^{t-1} \nabla f(x_s)\right\|^2\right] + \gamma^2 t \rho^2 + (t-1)t(t+1)\gamma^3 L \rho^2. \qquad (13)$$

Next, by Jensen's inequality and Lemma C.5,

$$\mathbb{E}\left[\left\|\sum_{s=0}^{t-1} \nabla f(x_s)\right\|^2\right] \le t \sum_{s=0}^{t-1} \mathbb{E}\left[\|\nabla f(x_s)\|^2\right]$$

$$\le t \sum_{s=0}^{t-1} \left(\|\nabla f(x_0)\|^2 + 2s\gamma L \rho^2\right)$$

$$\le t^2 \|\nabla f(x_0)\|^2 + (t-1)t(t+1)\gamma L \rho^2$$

where the last inequality uses $\sum_{s=0}^{t-1} 2s = t(t-1) \le (t-1)(t+1)$. Applying the above inequality to (13) we obtain the desired result. $\qquad \square$

### C.4 Proofs of Lemmas C.3 and C.2

*Proof of Lemma C.3.* Observe that given $\mathbf{x}_{\tau p}$, the sequence $x^i_{\tau p}, \ldots, x^i_{\tau(p+1)}$ is a sequence generated via stochastic gradient descent

$$x^i_{j+1} = x^i_j - \gamma \nabla f_{i,\xi^i_j}(x^i_j; x^{-i}_{\tau p})$$

for the $L_i$-smooth convex function $f_i(\cdot; x^{-i}_{\tau p})$, with $x^i_{\tau p}$ as initial point, using the stochastic oracle $\nabla f_{i,\xi^i}(\cdot; x^{-i}_{\tau p})$ satisfying *(BV)* (unbiased estimator of $\nabla f_i(\cdot; x^{-i}_{\tau p})$ with variance at most $\sigma_i^2$). Therefore, we can apply Lemma C.5 with

$$f(\cdot) = f_i(\cdot; x^{-i}_{\tau p}), \qquad \rho^2 = \sigma_i^2, \qquad x_0 = x^i_{\tau p}, \qquad x_s = x^i_j, \qquad L = L_i$$

and this immediately proves the desired statement. (Note that $s$ is replaced with $j - \tau p$ because $x^i_j$ is obtained by $j - \tau p$ steps of SGD from $x^i_{\tau p}$.)

$\qquad \square$

*Proof of Lemma C.2.* This is a direct consequence of Lemma C.9 with same choice of $f, \rho^2, x_0$ and $L$ as in the proof of Lemma C.3 and $x_s = x^i_{\tau p + t}$.

$\qquad \square$

## C.5 Remaining details in proof of Theorem 3.4

Note that the step-size condition of Lemma C.2 is satisfied by our step-size selection, as $\gamma < \frac{2}{\ell\tau + 2(\tau-1)L_{\max}\sqrt{\kappa}} \leq \frac{1}{L_{\max}(\tau-1)}$ (because $\kappa \geq 1$). Now combine Lemmas C.1 and C.2 to obtain

$$\mathbb{E}\left[\left\|\mathbf{x}_{\tau(p+1)} - \mathbf{x}_\star\right\|^2 \Big| \mathbf{x}_{\tau p}\right]$$

$$\leq (1 + \alpha\gamma(\tau-1))\left\|\mathbf{x}_{\tau p} - \mathbf{x}_\star\right\|^2 - 2\gamma\tau\left\langle \mathbf{x}_{\tau p} - \mathbf{x}_\star, \mathbf{F}(\mathbf{x}_{\tau p})\right\rangle$$

$$+ \sum_{j=\tau p+1}^{\tau(p+1)-1}\sum_{i=1}^{n}\frac{\gamma L_i^2}{\alpha}\left(\gamma^2(j-\tau p)^2\left\|\nabla f(x_{\tau p}^i; x_{\tau p}^{-i})\right\|^2 + \gamma^2(j-\tau p)\left(1 + 2(j-\tau p - 1)(j-\tau p+1)\gamma L_i\right)\sigma_i^2\right)$$

$$+ \sum_{i=1}^{n}\left(\gamma^2\tau^2\left\|\nabla f(x_{\tau p}^i; x_{\tau p}^{-i})\right\|^2 + \gamma^2\tau\left(1 + 2(\tau-1)(\tau+1)\gamma L_i\right)\sigma_i^2\right)$$

$$\leq (1 + \alpha\gamma(\tau-1))\left\|\mathbf{x}_{\tau p} - \mathbf{x}_\star\right\|^2 - 2\gamma\tau\left\langle \mathbf{x}_{\tau p} - \mathbf{x}_\star, \mathbf{F}(\mathbf{x}_{\tau p})\right\rangle + \left(\gamma^2\tau^2 + \frac{\gamma^3 L_{\max}^2\tau^2(\tau-1)}{3\alpha}\right)\left\|\mathbf{F}(\mathbf{x}_{\tau p})\right\|^2$$

$$+ \gamma^2\tau\left(1 + (\tau-1)\gamma L_{\max}\left(2(\tau+1) + \frac{L_{\max}}{2\alpha} + \frac{\gamma L_{\max}^2}{2\alpha}(\tau+1)^2\right)\right)\sigma^2$$

$$\tag{14}$$

where for the last inequality, we replace all occurrences of $L_i$'s by $L_{\max} = \max\{L_1, \dots, L_n\}$ and use the identities

$$\sigma^2 = \sum_{i=1}^{n}\sigma_i^2, \quad \left\|\mathbf{F}(\mathbf{x}_{\tau p})\right\|^2 = \sum_{i=1}^{n}\left\|\nabla f_i(x_{\tau p}^i; x_{\tau p}^{-i})\right\|^2$$

to eliminate the summations $\sum_{i=1}^{n}$ and use the following elementary summation results:

$$\sum_{j=\tau p+1}^{\tau(p+1)-1}(j-\tau p)^2 = \frac{(\tau-1)\tau(2\tau-1)}{6} \leq \frac{(\tau-1)\tau^2}{3}$$

$$\sum_{j=\tau p+1}^{\tau(p+1)-1}(j-\tau p) = \frac{(\tau-1)\tau}{2}$$

and

$$\sum_{j=\tau p+1}^{\tau(p+1)-1}(j-\tau p-1)(j-\tau p)(j-\tau p+1) = \frac{(\tau-2)(\tau-1)\tau(\tau+1)}{2} \leq \frac{(\tau-1)\tau(\tau+1)^2}{2}.$$

Now in (14), we use the assumption *(SCO)* to bound

$$-2\gamma\tau\left\langle \mathbf{x}_{\tau p} - \mathbf{x}_\star, \mathbf{F}(\mathbf{x}_{\tau p})\right\rangle + \left(\gamma^2\tau^2 + \frac{\gamma^3 L_{\max}^2\tau^2(\tau-1)}{3\alpha}\right)\left\|\mathbf{F}(\mathbf{x}_{\tau p})\right\|^2$$

$$\leq -\left(2\gamma\tau - \ell\left(\gamma^2\tau^2 + \frac{\gamma^3 L_{\max}^2\tau^2(\tau-1)}{3\alpha}\right)\right)\left\langle \mathbf{x}_{\tau p} - \mathbf{x}_\star, \mathbf{F}(\mathbf{x}_{\tau p})\right\rangle$$

$$= -\gamma\tau\left(2 - \gamma\ell\tau - \frac{\gamma^2\ell L_{\max}^2\tau(\tau-1)}{3\alpha}\right)\left\langle \mathbf{x}_{\tau p} - \mathbf{x}_\star, \mathbf{F}(\mathbf{x}_{\tau p})\right\rangle. \tag{15}$$

Provided that

$$2 - \gamma\ell\tau - \frac{\gamma^2\ell L_{\max}^2\tau(\tau-1)}{3\alpha} \geq 0, \tag{16}$$

we can again upper bound (15) using the assumption *(QSM)*:

$$-\gamma\tau\left(2 - \gamma\ell\tau - \frac{\gamma^2\ell L_{\max}^2\tau(\tau-1)}{3\alpha}\right)\left\langle \mathbf{x}_{\tau p} - \mathbf{x}_\star, \mathbf{F}(\mathbf{x}_{\tau p})\right\rangle$$

$$\leq -\gamma\tau\left(2 - \gamma\ell\tau - \frac{\gamma^2\ell L_{\max}^2\tau(\tau-1)}{3\alpha}\right)\mu\left\|\mathbf{x}_{\tau p} - \mathbf{x}_\star\right\|^2.$$

We plug this into (14) and rearrange the terms to obtain

$$
\mathbb{E}\left[\left\|\mathbf{x}_{\tau(p+1)} - \mathbf{x}_\star\right\|^2 \,\Big|\, \mathbf{x}_{\tau p}\right]
$$
$$
\leq \left(1 + \alpha\gamma(\tau-1) - \gamma\tau\left(2 - \gamma\tau\ell - \frac{\gamma^2\ell L_{\max}^2\tau(\tau-1)}{3\alpha}\right)\mu\right)\left\|\mathbf{x}_{\tau p} - \mathbf{x}_\star\right\|^2
$$
$$
+ \gamma^2\tau\left(1 + (\tau-1)\gamma L_{\max}\left(2(\tau+1) + \frac{L_{\max}}{2\alpha} + \frac{\gamma L_{\max}^2}{2\alpha}(\tau+1)^2\right)\right)\sigma^2. \tag{17}
$$

Now, we optimize the coefficient of the $\|\mathbf{x}_{\tau p} - \mathbf{x}_\star\|^2$ term in (17) by taking

$$
\alpha = \underset{\alpha>0}{\operatorname{argmin}}\, \alpha\gamma(\tau-1) + \frac{\gamma^3\ell L_{\max}^2\tau^2(\tau-1)\mu}{3\alpha} = \gamma\tau L_{\max}\sqrt{\frac{\ell\mu}{3}}.
$$

With this choice of $\alpha$, the bound (17) becomes

$$
\mathbb{E}\left[\left\|\mathbf{x}_{\tau(p+1)} - \mathbf{x}_\star\right\|^2 \,\Big|\, \mathbf{x}_{\tau p}\right]
$$
$$
\leq \left(1 - \gamma\tau\mu\left(2 - \gamma\ell\tau - 2(\tau-1)\gamma L_{\max}\sqrt{\frac{\ell}{3\mu}}\right)\right)\left\|\mathbf{x}_{\tau p} - \mathbf{x}_\star\right\|^2
$$
$$
+ \gamma^2\tau\left(1 + (\tau-1)\gamma L_{\max}\left(2(\tau+1) + \frac{1}{2\gamma\tau\sqrt{\ell\mu/3}} + \frac{L_{\max}(\tau+1)^2}{2\tau\sqrt{\ell\mu/3}}\right)\right)\sigma^2
$$
$$
\leq (1 - \gamma\tau\mu\zeta)\left\|\mathbf{x}_{\tau p} - \mathbf{x}_\star\right\|^2 + \gamma^2\tau\sigma^2\left(1 + (\tau-1)\left(4\gamma\tau L_{\max} + \frac{L_{\max}}{2\tau\sqrt{\ell\mu/3}} + \frac{\gamma\tau L_{\max}^2}{\sqrt{\ell\mu/3}}\right)\right) \tag{18}
$$

where for the last inequality, we use $\tau + 1 \leq 2\tau$ and make the substitution

$$
\zeta = 2 - \gamma\ell\tau - 2(\tau-1)\gamma L_{\max}\sqrt{\frac{\ell}{3\mu}} = 2 - \gamma\ell\tau - 2(\tau-1)\gamma L_{\max}\sqrt{\kappa/3}.
$$

Note that with our choice $\alpha = \gamma\tau L_{\max}\sqrt{\frac{\ell\mu}{3}}$ and $0 < \gamma < \frac{2}{\ell\tau + 2(\tau-1)L_{\max}\sqrt{\kappa}}$, the condition (16) is satisfied because

$$
2 - \gamma\ell\tau - \frac{\gamma^2\ell L_{\max}^2\tau(\tau-1)}{3\alpha} \geq 2 - \gamma\ell\tau - \frac{\gamma^2\ell L_{\max}^2\tau(\tau-1)}{3\alpha}
$$
$$
= 2 - \gamma\ell\tau - (\tau-1)\gamma L_{\max}\sqrt{\frac{\ell}{3\mu}}
$$
$$
\geq 2 - \gamma\left(\ell\tau + (\tau-1)L_{\max}\sqrt{\kappa}\right) > 0.
$$

Finally, unrolling the recursion (18) using the following simple lemma, with $a_p = \mathbb{E}\left[\|\mathbf{x}_{\tau p} - \mathbf{x}_\star\|^2\right]$, $A = \tau\mu\zeta$ and

$$
B = \tau\sigma^2\left(1 + (\tau-1)\left(4\gamma\tau L_{\max} + \frac{L_{\max}}{2\tau\sqrt{\ell\mu/3}} + \frac{\gamma\tau L_{\max}^2}{\sqrt{\ell\mu/3}}\right)\right)
$$

gives the desired rate. (Note that $\gamma A = \gamma\tau\mu\zeta \leq \gamma\tau\mu(2 - \gamma\ell\tau) \leq \gamma\ell\tau(2 - \gamma\ell\tau) \leq 1$.)

**Lemma C.10.** Let $\gamma, A, B > 0$ with $\gamma A \leq 1$. If a sequence $a_0, \ldots, a_R \in \mathbb{R}$ satisfies
$$
a_{p+1} \leq (1 - \gamma A)a_p + \gamma^2 B
$$
for $p = 0, \ldots, R-1$, then $a_R \leq (1 - \gamma A)^R a_0 + \frac{\gamma B}{A}$.

*Proof of Lemma C.10.* As there is nothing to prove if $\gamma A = 1$, suppose $\gamma A < 1$. Recursively applying the given inequality we have

$$a_R \leq (1 - \gamma A)a_{R-1} + \gamma^2 B \leq \cdots \leq (1 - \gamma A)^R a_0 + \gamma^2 B \sum_{p=0}^{R-1} (1 - \gamma A)^p.$$

Now apply the bound $\sum_{p=0}^{R-1}(1 - \gamma A)^p \leq \sum_{p=0}^{\infty}(1 - \gamma A)^p = \frac{1}{1-(1-\gamma A)} = \frac{1}{\gamma A}$ to the above inequality. $\qquad\square$

## C.6 Proof of Corollary 3.5

First, because $\eta > \kappa\tau$, we have

$$\gamma < \frac{1}{\mu\kappa\tau\left(1 + \frac{2L_{\max}}{\sqrt{\ell\mu}}\right)} = \frac{1}{\ell\tau\left(1 + \frac{2L_{\max}}{\sqrt{\ell\mu}}\right)} \leq \frac{1}{\ell\tau + 2(\tau - 1)L_{\max}\sqrt{\frac{\ell}{\mu}}} = \frac{1}{\ell\tau + 2(\tau - 1)L_{\max}\sqrt{\kappa}}.$$

Hence we can apply Theorem 3.4. Now observe that $\zeta > 2 - \gamma\left(\ell\tau + 2(\tau - 1)L_{\max}\sqrt{\kappa}\right) > 1$, and $(1 - u)^R \leq e^{-uR}$ for $u < 1$, so

$$(1 - \gamma\tau\mu\zeta)^R \leq e^{-\gamma\mu\zeta\tau R} \leq e^{-\gamma\mu T} = e^{-2\log\eta} = \frac{1}{\eta^2} = \frac{4(\log\eta)^2(1 + 2q)^2}{T^2} = \tilde{\mathcal{O}}\left(\frac{(1 + q)^2}{T^2}\right)$$

where we use $T = 2(1 + 2q)\eta\log\eta$ and remove the factor $\log\eta < \log T$ within the $\tilde{\mathcal{O}}$ notation. Next, for the terms proportional to $\sigma^2$, we have

$$\left(1 + (\tau - 1)\left(4\gamma\tau L_{\max} + \frac{L_{\max}}{2\tau\sqrt{\ell\mu/3}} + \frac{\gamma\tau L_{\max}^2}{\sqrt{\ell\mu/3}}\right)\right)\frac{\gamma\sigma^2}{\mu\zeta}$$

$$\leq \frac{\gamma\sigma^2}{\mu}\left(1 + \tau\left(4\gamma\tau L_{\max} + \frac{\sqrt{3}q}{2\tau} + \sqrt{3}\gamma\tau L_{\max}q\right)\right)$$

$$\leq \frac{\gamma\sigma^2}{\mu}\left(1 + \frac{\sqrt{3}q}{2}\right) + \frac{\gamma^2\tau^2 L_{\max}\sigma^2}{\mu}(4 + \sqrt{3}q)$$

$$= \frac{\sigma^2(1 + \sqrt{3}q/2)}{\mu^2\eta(1 + 2q)} + \frac{\tau^2 L_{\max}\sigma^2(4 + \sqrt{3}q)}{\mu^3\eta^2(1 + 2q)^2}$$

$$= \tilde{\mathcal{O}}\left(\frac{(1 + q)\sigma^2}{\mu^2 T} + \frac{(1 + q)\tau^2 L_{\max}\sigma^2}{\mu^3 T^2}\right).$$

Combining these with Theorem 3.4 we arrive at the desired conclusion.

## C.7 Proof of Theorem 3.6

Note that we use constant step-size $\gamma_k \equiv \gamma_{\tau p}$ within each communication round $p$, i.e., for $\tau p \leq k \leq \tau(p + 1) - 1$, so we can apply the bound (18) from the proof of Theorem 3.4, provided that

$$\gamma_{\tau p} \leq \frac{1}{\ell\tau + 2(\tau - 1)L_{\max}\sqrt{\kappa}}.$$

This clearly holds true when $p < 2(1 + 2q)\kappa - 1$, and when $p \geq 2(1 + 2q)\kappa - 1$ then

$$\gamma_{\tau p} = \frac{1}{\tau\mu}\frac{2p + 1}{(p + 1)^2} < \frac{1}{\tau\mu}\frac{2}{p + 1} \leq \frac{1}{\tau\mu}\frac{1}{(1 + 2q)\kappa} = \frac{1}{\ell\tau + 2\tau L_{\max}\sqrt{\kappa}}$$

so we see that the step-size condition is satisfied. Furthermore we have

$$\zeta_{\tau p} = 2 - \gamma_{\tau p}\ell\tau - 2(\tau - 1)\gamma_{\tau p}L_{\max}\sqrt{\kappa/3} > 1,$$

so (18), with $q = \frac{L_{\max}}{\sqrt{\ell\mu}}$ and taking expectation with respect to $\mathbf{x}_{\tau p}$, gives

$$\mathbb{E}\left[\left\|\mathbf{x}_{\tau(p+1)} - \mathbf{x}_\star\right\|^2\right] \leq (1 - \gamma_{\tau p}\tau\mu\zeta_{\tau p})\mathbb{E}\left[\left\|\mathbf{x}_{\tau p} - \mathbf{x}_\star\right\|^2\right]$$

$$+ \gamma_{\tau p}^2\tau\sigma^2\left(1 + (\tau - 1)\left(\gamma_{\tau p}\tau L_{\max}(4 + \sqrt{3}q) + \frac{\sqrt{3}}{2\tau q}\right)\right)$$

$$\leq (1 - \gamma_{\tau p}\tau\mu)\mathbb{E}\left[\left\|\mathbf{x}_{\tau p} - \mathbf{x}_\star\right\|^2\right] + (1 + q)\gamma_{\tau p}^2\tau\sigma^2 + 4(1 + q)\gamma_{\tau p}^3\tau^2(\tau - 1)L_{\max}\sigma^2.$$

$$(19)$$

For $p \geq 2(1 + 2q)\kappa - 1$, plugging in $\gamma_{\tau p} = \frac{1}{\tau\mu}\frac{2p+1}{(p+1)^2}$ we obtain

$$\mathbb{E}\left[\left\|\mathbf{x}_{\tau(p+1)} - \mathbf{x}_\star\right\|^2\right] \leq \frac{p^2}{(p+1)^2}\mathbb{E}\left[\left\|\mathbf{x}_{\tau p} - \mathbf{x}_\star\right\|^2\right] + \frac{(2p+1)^2\sigma^2(1+q)}{\tau\mu^2(p+1)^4}\left(1 + \frac{4(\tau - 1)L_{\max}(2p+1)}{\mu(p+1)^2}\right).$$

Multiplying $\tau^2(p + 1)^2$ to both sides and upper-bounding $\frac{2p+1}{p+1} \leq 2$, we obtain

$$(\tau(p + 1))^2\mathbb{E}\left[\left\|\mathbf{x}_{\tau(p+1)} - \mathbf{x}_\star\right\|^2\right] \leq (\tau p)^2\mathbb{E}\left[\left\|\mathbf{x}_{\tau p} - \mathbf{x}_\star\right\|^2\right] + \frac{4(1 + q)\tau\sigma^2}{\mu^2}\left(1 + \frac{8(\tau - 1)L_{\max}}{\mu(p + 1)}\right).$$

Let $p_0 = \lceil 2(1 + 2q)\kappa - 1\rceil$. Chaining the above inequality for $p = p_0, \ldots, R - 1$ gives

$$(\tau R)^2\mathbb{E}\left[\left\|\mathbf{x}_{\tau R} - \mathbf{x}_\star\right\|^2\right]$$

$$\leq (\tau p_0)^2\mathbb{E}\left[\left\|\mathbf{x}_{\tau p_0} - \mathbf{x}_\star\right\|^2\right] + \frac{4(1 + q)\tau(R - p_0)\sigma^2}{\mu^2} + \frac{32(1 + q)\tau(\tau - 1)L_{\max}\sigma^2}{\mu^3}\sum_{p=p_0}^{R-1}\frac{1}{p + 1}$$

$$\leq (\tau p_0)^2\mathbb{E}\left[\left\|\mathbf{x}_{\tau p_0} - \mathbf{x}_\star\right\|^2\right] + \frac{4(1 + q)\tau(R - p_0)\sigma^2}{\mu^2} + \frac{32(1 + q)\tau^2 L_{\max}\sigma^2\log(R/p_0)}{\mu^3}$$

where we use $\sum_{p=p_0}^{R-1}\frac{1}{p+1} \leq \int_{p_0}^R\frac{dp}{p} = \log\frac{R}{p_0}$. Now substitute $T = \tau R$ using the upper bounds $\tau(R - p_0) \leq \tau R = T$ and $\log(R/p_0) \leq \log T$, we can write

$$T^2\mathbb{E}\left[\left\|\mathbf{x}_T - \mathbf{x}_\star\right\|^2\right] \leq (\tau p_0)^2\mathbb{E}\left[\left\|\mathbf{x}_{\tau p_0} - \mathbf{x}_\star\right\|^2\right] + \frac{4(1 + q)T\sigma^2}{\mu^2} + \frac{32(1 + q)\tau^2 L_{\max}\sigma^2\log T}{\mu^3}.$$

$$(20)$$

As $\gamma_k$ is constantly $\gamma_0 = \frac{1}{\ell\tau(1+2q)}$ over rounds $p = 0, \ldots, p_0 - 1$, we can directly apply Theorem 3.4 with $R = p_0$ and similar simplification of the $\sigma^2$-terms as in (19) to bound

$$\mathbb{E}\left[\left\|\mathbf{x}_{\tau p_0} - \mathbf{x}_\star\right\|^2\right] \leq \left(1 - \frac{\mu}{\ell(1 + 2q)}\right)^{p_0}\left\|\mathbf{x}_0 - \mathbf{x}_\star\right\|^2 + \frac{(1 + q)\gamma_0\sigma^2}{\mu}\left(1 + 4\gamma_0\tau(\tau - 1)L_{\max}\right)$$

$$\leq \left(1 - \frac{1}{\kappa(1 + 2q)}\right)^{\kappa(1+2q)}\left\|\mathbf{x}_0 - \mathbf{x}_\star\right\|^2 + \frac{\sigma^2}{\ell\mu\tau}\left(1 + \frac{4(\tau - 1)L_{\max}}{\ell(1 + 2q)}\right)$$

$$\leq \frac{\left\|\mathbf{x}_0 - \mathbf{x}_\star\right\|^2}{e} + \frac{\sigma^2}{\ell\mu\tau}\left(1 + \frac{2\tau}{\sqrt{\kappa}}\right),$$

where the second line uses $p_0 \geq 2(1 + 2q)\kappa - 1 \geq \kappa(1 + 2q)$, and the third line uses the bound $\left(1 - \frac{1}{t}\right)^t \leq \frac{1}{e}$ for $t > 1$ and $\frac{4(\tau-1)L_{\max}}{\ell(1+2q)} \leq \frac{4q\tau\sqrt{\ell\mu}}{\ell(1+2q)} \leq 2\tau\sqrt{\frac{\mu}{\ell}} = \frac{2\tau}{\sqrt{\kappa}}$. Now plugging this into (20) and dividing both sides by $T^2$ we obtain

$$\mathbb{E}\left[\left\|\mathbf{x}_T - \mathbf{x}_\star\right\|^2\right]$$

$$\leq \frac{p_0^2\tau^2\left\|\mathbf{x}_0 - \mathbf{x}_\star\right\|^2}{eT^2} + \frac{\tau p_0^2\sigma^2}{\ell\mu T^2}\left(1 + \frac{2\tau}{\sqrt{\kappa}}\right) + \frac{4(1 + q)\sigma^2}{\mu^2 T} + \frac{32(1 + q)\tau^2 L_{\max}\sigma^2\log T}{\mu^3 T^2}$$

$$\leq \frac{4(1 + 2q)^2\kappa^2\tau^2\left\|\mathbf{x}_0 - \mathbf{x}_\star\right\|^2}{eT^2} + \frac{4(1 + q)\sigma^2}{\mu^2 T} + \frac{4(1 + 2q)^2\kappa\tau\sigma^2}{\mu^2 T^2}\left(1 + \frac{2\tau}{\sqrt{\kappa}}\right) + \frac{32(1 + q)\tau^2 L_{\max}\sigma^2\log T}{\mu^3 T^2}.$$

which is the desired result.

# D   Details of Numerical Experiments

Experiments were conducted using a personal MacBook with an Apple M3 chip and 16GB RAM.

## D.1   Quadratic $n$-player game

We set $n = 5$, $d = 10$ and $M = 100$. The matrices $\mathbf{A}_{i,m}$ are generated randomly with their eigenvalues in the range $[\mu_{\mathbf{A}}, L_{\mathbf{A}}]$ ($0 < \mu_{\mathbf{A}} < L_{\mathbf{A}}$). Similarly, for $1 \leq i < j \leq n$, we generate the matrices $\mathbf{B}_{i,j,m}$ randomly with their eigenvalues in $[0, L_{\mathbf{B}}]$. Notably, we set $\mathbf{B}_{j,i,m} = -\mathbf{B}_{i,j,m}^{\mathsf{T}}$ for $1 \leq j < i \leq n$. With this condition, we can ensure that the $n$-player game (2) satisfies the *(QSM)* assumption, regardless of the values of $\mu_{\mathbf{A}}, L_{\mathbf{A}}$ and $L_{\mathbf{B}}$. We show below why this is the case.

Recall that we have

$$f_i(x^1, \ldots, x^n) = \frac{1}{2} \langle x^i, \mathbf{A}_i x^i \rangle + \langle a_i, x^i \rangle + \sum_{j \neq i} \langle x^i, \mathbf{B}_{i,j} x^j \rangle$$

for $i = 1, \ldots, n$. Differentiating $f_i$ with respect to $x^i$, we get

$$\nabla f_i(x^i; x^{-i}) = \mathbf{A}_i x^i + a_i + \sum_{j \neq i} \mathbf{B}_{i,j} x^j$$

and thus

$$\nabla f_i(x^i; x^{-i}) - \nabla f_i(x_\star^i; x_\star^{-i}) = \left( \mathbf{A}_i x^i + a_i + \sum_{j \neq i} \mathbf{B}_{i,j} x^j \right) - \left( \mathbf{A}_i x_\star^i + a_i + \sum_{j \neq i} \mathbf{B}_{i,j} x_\star^j \right)$$

$$= \mathbf{A}_i(x^i - x_\star^i) + \sum_{j \neq i} \mathbf{B}_{i,j}(x^j - x_\star^j)$$

and

$$\langle \mathbb{F}(\mathbf{x}) - \mathbb{F}(\mathbf{x}_\star), \mathbf{x} - \mathbf{x}_\star \rangle = \sum_{i=1}^n \left\langle \nabla f_i(x^i; x^{-i}) - \nabla f_i(x_\star^i; x_\star^{-i}), x^i - x_\star^i \right\rangle$$

$$= \sum_{i=1}^n \left\langle x^i - x_\star^i, \mathbf{A}_i(x^i - x_\star^i) \right\rangle + \sum_{i=1}^n \sum_{j \neq i} \left\langle x^i - x_\star^i, \mathbf{B}_{i,j}(x^j - x_\star^j) \right\rangle.$$

Now, the double summation term vanishes because for any $i \neq j$,

$$\left\langle x^i - x_\star^i, \mathbf{B}_{i,j}(x^j - x_\star^j) \right\rangle + \left\langle x^j - x_\star^j, \mathbf{B}_{j,i}(x^i - x_\star^i) \right\rangle = 0$$

due to the condition $\mathbf{B}_{j,i} = -\mathbf{B}_{i,j}^{\mathsf{T}}$. Therefore, provided that each $\mathbf{A}_i \succeq \mu I$ we see that $\mathbb{F}$ satisfies *(QSM)* (in fact, the same argument with arbitrary $\mathbf{y}$ in place of $\mathbf{x}_\star$ shows that $\mathbb{F}$ is $\mu$-strongly monotone).

## D.2   Distributed mobile robot control

We follow the same choice of parameter values $a_i, b_i, x_{\mathrm{anc}}^i, h_{ij}$ within (3) from [60]: $n = 5$, $d = 1$, $a_i = 10 + i/6$, $b_i = i/6$,

$$\left( x_{\mathrm{anc}}^1, x_{\mathrm{anc}}^2, x_{\mathrm{anc}}^3, x_{\mathrm{anc}}^4, x_{\mathrm{anc}}^5 \right) = (1, -4, 8, -9, 13)$$

and

$$(h_{ij})_{\substack{1 \leq i \leq 5 \\ 1 \leq j \leq 5}} = \begin{bmatrix} 0 & 5 & -7 & 9 & -8 \\ -5 & 0 & -6 & 2 & -9 \\ 7 & 6 & 0 & 7 & -4 \\ -9 & -2 & -7 & 0 & -2 \\ 8 & 9 & 4 & 2 & 0 \end{bmatrix}.$$

We add Gaussian noise with $\sigma^2 = 100$ to the gradients to simulate stochasticity. In this setup, all our theoretical assumptions are satisfied [60].

# E  Additional Experiments

## E.1  Quadratic $n$-player game with step-size tuning

In this experiment, we simulate the scenario where we do not know the precise theoretical parameters in advance. For each $\tau \in \{1, 2, 4, 5, 8, 20\}$, we tune $\gamma$ by running PEARL-SGD with each $\gamma \in \{10^{-1}, 10^{-2}, \dots, 10^{-6}\}$, and plot the best relative error $\|\mathbf{x}_{\tau p} - \mathbf{x}_\star\|^2 / \|\mathbf{x}_0 - \mathbf{x}_\star\|^2$ ($y$-axis) versus the communication round index $p$ ($x$-axis). Figure 5a presents results from Deterministic PEARL-SGD, and Figure 5b presents results under stochasticity, imposed by mini-batching from the finite sum. The results demonstrate that in practice, we can use $(\tau, \gamma)$ as tunable hyperparameters to achieve the best communication complexity.

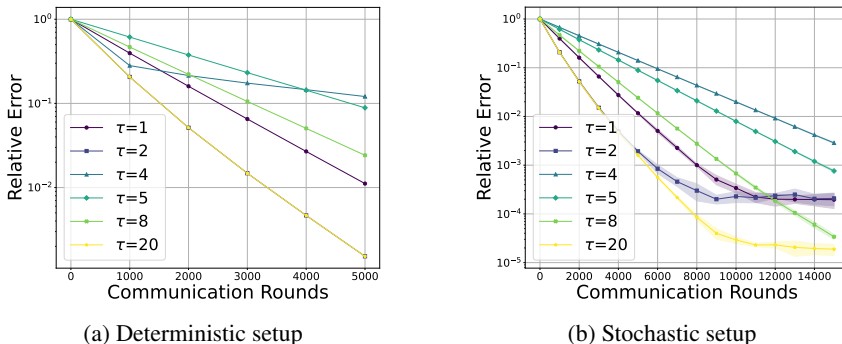

(a) Deterministic setup
(b) Stochastic setup

Figure 5: Performance plots for PEARL-SGD on the $n$-player game (2) with different values of $\tau$. For each $\tau$, we use the empirically tuned step-size $\gamma \in \{10^{-1}, 10^{-2}, \dots, 10^{-6}\}$ for the best relative error $\frac{\|\mathbf{x}_{\tau p} - \mathbf{x}_\star\|^2}{\|\mathbf{x}_0 - \mathbf{x}_\star\|^2}$. Figure 5a shows the result from deterministic setup and 5b shows the stochastic setup.

## E.2  Distributed mobile robot control

In Fig. 6, we plot the local objective values $f_i$ in the setup (3) obtained from PEARL-SGD for each robot (player) $i = 1, \dots, 5$ in the case $\tau = 5$. Generally in games, the objectives $f_i$ can have both cooperative and competitive components. After the cooperative components in each $f_i$ are sufficiently reduced, $f_i$'s can oscillate due to the competing interests of players until an equilibrium is found, and then finally stabilize around a certain level.

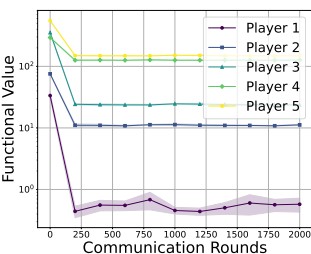

Figure 6: Local objective $f_i$ in mobile robot control setup.

# F Discussion on Theoretical Assumptions

## F.1 Possible simplification of assumptions: Assuming cocoercivity of $\mathbb{F}$

In fact, the convergence of PEARL-SGD can still be proved even if the three assumptions *(CVX)*, *(SM)* and *(SCO)* are replaced with the single assumption that $\mathbb{F}: \mathbb{R}^D \to \mathbb{R}^D$ is $\frac{1}{\ell}$-cocoercive, i.e.,

$$\langle \mathbb{F}(\mathbf{x}) - \mathbb{F}(\mathbf{y}), \mathbf{x} - \mathbf{y} \rangle \geq \frac{1}{\ell} \|\mathbf{x} - \mathbf{y}\|^2, \quad \forall \mathbf{x}, \mathbf{y} \in \mathbb{R}^D. \tag{\textit{COCO}}$$

In the subsequent paragraphs, we explain in detail why this is the case. However, we emphasize here that if we derived all convergence theory using *(COCO)* in place of *(CVX)*, *(SM)* and *(SCO)* and did not distinguish the role of $L_i$'s (the local Lipschitzness parameters from *(SM)*) from that of $\ell$, then the resulting convergence rates would have become much more pessimistic (worse) in many cases. Therefore, in our work, we choose to use the current set of assumptions. It allows us to more clearly present the tight dependency of convergence rates to $L_i$'s. Also note that assuming *(CVX)*, *(SM)* and *(SCO)* is strictly more general than assuming *(COCO)*, as we illustrate in Appendix F.2.

**_(COCO)_ implies *(CVX)*, *(SM)* and *(SCO)*.** Trivially, *(COCO)* implies *(SCO)*. Furthermore, if $\mathbb{F}$ is $\frac{1}{\ell}$-cocoercive, then $\mathbb{F}$ is monotone:

$$\langle \mathbb{F}(\mathbf{x}) - \mathbb{F}(\mathbf{y}), \mathbf{x} - \mathbf{y} \rangle \geq 0, \quad \forall \mathbf{x}, \mathbf{y} \in \mathbb{R}^D, \tag{21}$$

and $\ell$-Lipschitz continuous:

$$\|\mathbb{F}(\mathbf{x}) - \mathbb{F}(\mathbf{y})\| \leq \ell \|\mathbf{x} - \mathbf{y}\|, \quad \forall \mathbf{x}, \mathbf{y} \in \mathbb{R}^D. \tag{22}$$

In particular, for each $i = 1, \ldots, n$, we can take

$$\mathbf{x} = (x^1, \ldots, x^{i-1}, x^i, x^{i+1}, \ldots, x^n), \quad \mathbf{y} = (x^1, \ldots, x^{i-1}, y^i, x^{i+1}, \ldots, x^n) \tag{23}$$

in (21), which gives

$$\langle \nabla f_i(x^i; x^{-i}) - \nabla f_i(y^i; x^{-i}), x^i - y^i \rangle \geq 0$$

for any $x^i, y^i \in \mathbb{R}^{d_i}$ and $x^{-i} \in \mathbb{R}^{D-d_i}$. That is, the gradient of $f_i(\cdot; x^{-i}): \mathbb{R}^{d_i} \to \mathbb{R}$ is a monotone operator on $\mathbb{R}^{d_i}$, and this implies that $f_i(\cdot; x^{-i})$ is convex, i.e., *(CVX)* holds. Similarly, plugging the choice (23) into (22) we obtain

$$\|\nabla f_i(x^i; x^{-i}) - \nabla f_i(y^i; x^{-i})\| \leq \ell \|x^i - y^i\|,$$

showing that *(SM)* holds, with $L_i = \ell$. Therefore, all theorems from the main paper hold under the assumptions *(QSM)*, *(COCO)*, and *(BV)*, with $\ell$ in place of $L_{\max}$ in step-size restrictions and convergence rates.

**What do we lose by replacing $L_{\max}$ with $\ell$?** The previous discussion shows that we can assume *(COCO)* and replace all occurrences of $L_{\max}$ with $\ell$ within the theory. In this case, however, the step-size conditions in Theorems 3.3 and 3.4 become

$$\gamma \leq \frac{1}{\ell(\tau + 2(\tau - 1)\sqrt{\kappa})} = \mathcal{O}\left(\frac{1}{\ell\tau\sqrt{\kappa}}\right), \tag{24}$$

and the $\sqrt{\kappa}$ factor in the denominator is undesirable as it significantly restricts the range of step-size one can use if $\kappa$ is large. Furthermore, in Corollary 3.5 and Theorem 3.6, the factor $q$ becomes $\sqrt{\frac{\ell}{\mu}} = \sqrt{\kappa}$, causing the constant factors in the convergence bounds to potentially become large.

However, there are many cases where $L_{\max} \ll \ell$, showing why it is beneficial to keep the dependency on $L_{\max}$ tight as we do. As an abstract example, when $\mathbb{F}$ is a generic $\mu$-strongly monotone and $M$-Lipschitz continuous operator, the tight (smallest) cocoercivity parameter one can guarantee on $\mathbb{F}$ is $\ell = M^2/\mu$ [33] (tightness can be shown using, e.g., the scaled relative graph theory in [115], [114, Chapter 13]). On the other hand, we have

$$L_{\max} \leq \max_{i=1,\ldots,n} \sup_{\substack{\mathbf{x}=(x^i,x^{-i}), \mathbf{y}=(y^i,x^{-i}) \\ x^i \neq y^i}} \frac{\|\mathbb{F}(\mathbf{x}) - \mathbb{F}(\mathbf{y})\|}{\|\mathbf{x} - \mathbf{y}\|} \leq \sup_{\mathbf{x} \neq \mathbf{y}} \frac{\|\mathbb{F}(\mathbf{x}) - \mathbb{F}(\mathbf{y})\|}{\|\mathbf{x} - \mathbf{y}\|} = M,$$

i.e., $M$ is an upper bound on $L_{\max}$ (better than $\ell$). Therefore, $\ell$ is at least $\frac{\ell}{M} = \frac{\ell}{\sqrt{\ell\mu}} = \sqrt{\kappa}$ times larger than $L_{\max}$, and the largest step-size allowed in Theorems 3.3 and 3.4 is

$$\frac{1}{\ell\tau + 2(\tau-1)L_{\max}\sqrt{\kappa}} = \Omega\left(\frac{1}{\ell\tau}\right)$$

which is in contrast with (24) where we used $\ell$ in place of $L_{\max}$ and obtained $\sqrt{\kappa}$ times smaller step-size range. Additionally, note that in this case $q = \frac{L_{\max}}{\sqrt{\ell\mu}} = \frac{L_{\max}}{M} \leq 1$ in Corollary 3.5 and Theorem 3.6, so we can avoid the $\kappa$-dependent factors appearing in the convergence results.

We demonstrate another problem class for which $L_{\max} \ll \ell$. Consider a two-player matrix game, regularized by adding strongly convex (resp. strongly concave) quadratic terms in $x$ (resp. $y$):

$$\operatorname*{minimize}_{u\in\mathbb{R}^m} \operatorname*{maximize}_{v\in\mathbb{R}^m} \mathcal{L}(u,v) = \frac{\mu}{2}\|u\|^2 + g^\mathsf{T}u + u^\mathsf{T}\mathbf{B}v - h^\mathsf{T}v - \frac{\mu}{2}\|v\|^2 \tag{25}$$

where $\mathbf{B} \in \mathbb{R}^{m\times m}, g, h \in \mathbb{R}^m$. In our $n$-player game notation, the first and second players respectively use the objective function $f_1(x^1; x^2) = \mathcal{L}(x^1, x^2)$ and $f_2(x^2; x^1) = -\mathcal{L}(x^1, x^2)$. In this case, the operator $\mathbb{F}$ is $\mu$-strongly monotone with $\mu$ and $M$-Lipschitz continuous with parameter $M \geq \sqrt{\|\mathbf{B}\|_2^2 + \mu^2} \geq \|\mathbf{B}\|_2$. Note that the cocoercivity parameter $\ell$ is at least $M$ (and at most $M^2/\mu$). On the other hand,

$$\nabla f_1(x^1; x^2) = \mu x^1 + g + \mathbf{B}x^2, \quad \nabla f_2(x^2; x^1) = \mu x^2 + h - \mathbf{B}^\mathsf{T}x^1,$$

so the Lipschitz constant for $\nabla f_1$ with $x^2$ fixed (resp. $\nabla f_2$ with $x^1$ fixed) is $\mu$, i.e., $L_{\max} = \mu$. Therefore, we have $L_{\max} \ll \ell$ in this scenario, as strength of regularization $\mu$ is usually small compared to the smoothness parameter $M$. The same principle applies to the $n$-player analogue of this setup we use in Section 4.1, where each player has the objective function

$$f_i(x^i; x^{-i}) = \frac{1}{2}\left\langle x^i, \mathbf{A}_i x^i \right\rangle + \left\langle a_i, x^i \right\rangle + \sum_{\substack{1\leq j\leq n \\ j\neq i}} \left\langle x^i, \mathbf{B}_{i,j}x^j \right\rangle$$

with $\mathbf{B}_{j,i} = -\mathbf{B}_{i,j}^\mathsf{T}$. If the quadratic terms are the small regularization terms introduced to induce convergence, so that $\mathbf{A}_i = \mu\mathbf{I}$ with $\mu \ll \|\mathbf{B}_{i,j}\|_2$, then we have $L_{\max} = \mu \ll \max_{i\neq j}\|\mathbf{B}_{i,j}\|_2 \leq \ell$.

### F.2   Example of non-cocoercive $\mathbb{F}$ satisfying *(CVX)*, *(SM)*, *(QSM)* and *(SCO)*

Consider the two-player game where two players have the objectives

$$f_1(u; v) = \frac{u^2}{2}\varphi(v)$$

$$f_2(v; u) = \frac{v^2}{2}\varphi(u)$$

where $\varphi\colon \mathbb{R} \to \mathbb{R}$ is defined by

$$\varphi(t) = \left(\mu + (\ell - \mu)\sin^2 t\right).$$

Here $0 < \mu < \ell$, and we use the notation $\mathbf{x} = (u, v) \in \mathbb{R} \times \mathbb{R}$ instead of $\mathbf{x} = (x^1, x^2)$ for better readability. Note that because $\varphi$ satisfies

$$0 < \mu \leq \varphi(t) \leq \ell, \quad \forall t \in \mathbb{R},$$

$f_1(\cdot, v)\colon \mathbb{R} \to \mathbb{R}$ is convex (quadratic) for any $v \in \mathbb{R}$, and so is $f_2(u, \cdot)$ for any $u \in \mathbb{R}$. Therefore, this game satisfies *(CVX)*. For any $\mathbf{x} = (u, v)$, we have

$$\mathbb{F}(\mathbf{x}) = (\nabla_u f_1(u; v), \nabla_v f_2(v; u)) = (u\varphi(v), v\varphi(u)).$$

Therefore, the unique equilibrium of the game is $\mathbf{x}_\star = (u_\star, v_\star) = (0, 0)$. Additionally, observe that

$$\nabla_{uu}f_1(u; v) = \varphi(v) \in [\mu, \ell], \quad \nabla_{vv}f_2(v; u) = \varphi(u) \in [\mu, \ell].$$

In particular, the both second derivatives are bounded, so *(SM)* is satisfied. Next, we have

$$\langle \mathbb{F}(\mathbf{x}), \mathbf{x} - \mathbf{x}_\star \rangle = u^2\varphi(v) + v^2\varphi(u) \geq \mu(u^2 + v^2) = \mu\|\mathbf{x} - \mathbf{x}_\star\|^2,$$

i.e., $\mathbb{F}$ satisfies *(QSM)*. Finally, we have

$$\|\mathbb{F}(\mathbf{x})\|^2 = u^2\varphi(v)^2 + v^2\varphi(u)^2 \leq \max\{\varphi(v),\varphi(u)\}\left(u^2\varphi(v) + v^2\varphi(u)\right) \leq \ell\left\langle\mathbb{F}(\mathbf{x}), \mathbf{x} - \mathbf{x}_\star\right\rangle,$$

showing that $\mathbb{F}$ satisfies *(SCO)*.

On the other hand, $\mathbb{F}$ is not cocoercive with respect to any parameter; in fact, it is not even Lipschitz continuous nor monotone. Observe that the cross-derivatives

$$\nabla_{uv}f_1(u;v) = (\ell - \mu)u\sin(2v), \quad \nabla_{vu}f_2(u;v) = (\ell - \mu)v\sin(2u)$$

are unbounded over $\mathbb{R} \times \mathbb{R}$, so $\mathbb{F}$ is cannot be Lipschitz continuous with any fixed parameter. Furthermore, we have

$$\left(D\mathbb{F} + D\mathbb{F}^{\mathsf{T}}\right)(u,v) = \begin{bmatrix} 2\varphi(v) & (\ell - \mu)(u\sin(2v) + v\sin(2u)) \\ (\ell - \mu)(u\sin(2v) + v\sin(2u)) & 2\varphi(u) \end{bmatrix}$$

so with $u = v = \left(2N + \frac{1}{2}\right)\pi$, we have

$$\det\left(D\mathbb{F} + D\mathbb{F}^{\mathsf{T}}\right)(u,v) = 4\varphi^2\left(\left(2N + \frac{1}{2}\right)\pi\right) - 4(\ell - \mu)^2\left(2N + \frac{1}{2}\right)^2\pi^2$$

$$= 4\ell^2 - 4(\ell - \mu)^2\left(2N + \frac{1}{2}\right)^2\pi^2$$

$$< 0$$

provided that $N$ is sufficiently large. As a differentiable operator $\mathbb{F}$ is monotone if and only if $D\mathbb{F} + D\mathbb{F}^{\mathsf{T}} \succeq 0$ everywhere [114], this shows that $\mathbb{F}$ is not monotone.

Note that while we provided a two-player example for simplicity, one can easily use the essentially same ideas to construct a non-cocoercive $n$-player game satisfying our assumptions with any $n > 2$. For example, we can choose $f_i(x^i; x^{-i}) = \frac{(x^i)^2}{2}\varphi(x^{i+1})$ where we identify $x^{n+1} = x^1$.

