# OpenReview forum: "Multiplayer Federated Learning: Reaching Equilibrium with Less Communication"
_NeurIPS.cc/2025/Conference — NeurIPS 2025 poster_

### Official Review · Reviewer_jQKH · 2025-06-27

**Clarity:** 3
**Significance:** 2
**Originality:** 3
**Rating:** 4
**Confidence:** 3

**Summary:**

This paper studies federated learning from a game-theoretic perspective, modeling the clients as rational players with individual objectives and strategic behaviors. When every player optimizes their own utility function and observes other players' actions periodically, the system should reach an equilibrium. By proposing a per-player local SGD algorithm, the authors find that by doing local updates, the system would reach a neighborhood of the equilibrium with less communication under several strong assumptions: convexity, smoothness, bounded variance, quasi-strong monotonicity and star-cocoercivity. Numerical experiments are performed to verify the theory.

**Questions:**

1. In classical FL, all the clients have a collaborative goal, and intuitively, by leveraging information from other clients, they would get a better result than training on their own. But in the MpFL framework, clients no longer have collaboration in mind, or even that they would be competitive with each other, what would be the incentive for them to participate in the system if the training may harm their own benefits compared to working alone?
2. typo: line 212: etragradient -> extragradient
3. Figure 2(c) has a weird large shaded region in the bottom left.
4. Are all the experiments using a constant step size? Are there any numerical results to verify the sub-linear convergence to exact equilibrium in the stochastic case with decreasing step sizes?

**Ethical Concerns:**

["NO or VERY MINOR ethics concerns only"]

**Final Justification:**

This paper introduces a novel framework at the intersection of federated learning and game theory, proposing a per-player local SGD algorithm and analyzing its convergence to equilibrium under several strong assumptions. While the theoretical analysis relies on restrictive conditions and the experimental validation is limited to small-scale convex settings, the core idea and interdisciplinary framework are compelling. The rebuttal solved most of my questions, but the limitations of the work still exist. Given these strengths and limitations, I rate the paper as borderline accept.

**Limitations:**

yes

**Quality:**

3

**Strengths And Weaknesses:**

Strengths:
The motivation and the modeling of this work from a game-theoretic perspective are interesting, and the paper is generally well-written and easy to follow.

Weaknesses:
1. The MpFL framework is defined without a discussion on the existence of the solution. The goal of the MpFL problem is to find a pure strategy equilibrium, which does not necessarily exist, but there is no discussion in the paper under which necessary or sufficient conditions the problem is well-defined.
2. The assumptions in this paper are strong and restrictive. Only convex local loss functions are considered, while in the real-world applications, most cases are non-convex, but there is neither theoretical analysis nor numerical experiments on the non-convex case.

---

> ### Author Rebuttal · Authors · 2025-07-31
>
> We sincerely thank the reviewer for highlighting that our paper is interesting and well-written, for the constructive feedback, and for their time and effort spent reviewing. Below, we provide responses to each point.
>
> ## Weaknesses
>
>
> > 1. The MpFL framework is defined without a discussion on the existence of the solution. The goal of the MpFL problem is to find a pure strategy equilibrium, which does not necessarily exist, but there is no discussion in the paper under which necessary or sufficient conditions the problem is well-defined.
>
>
> We thank the reviewer for their careful reading. Under our assumptions, equilibrium always exists due to classical results (please see [1, Theorem 1] and references therein) as the minimization objectives $f\_i$ of individual players are continuous and convex in $x\_i$ — *equivalent to the payoff functions being concave*. We can easily add this information to the updated version of our work.
>
> [1] Daskalakis. Non-Concave Games: A Challenge for Game Theory’s Next 100 Years. 2022.
>
>
> > 2. The assumptions in this paper are strong and restrictive. Only convex local loss functions are considered, while in the real-world applications, most cases are non-convex, but there is neither theoretical analysis nor numerical experiments on the non-convex case.
>
> Our work is the first paper introducing the MpFL framework with the first algorithm (PEARL-SGD) for solving it. As such, we focus on a setting in which we can provide tight convergence guarantees, under the known most relaxed conditions for the convergence of player-wise SGD in the centralized (non-federated) multiplayer game setting. This parallels how numerous theoretical works in classical FL have been done ([2, 3, 4] to name a few), assuming strong convexity while primary applications are ML models.
>
> We believe that not all papers should be about nonconvex settings and deep neural networks. This will be, of course, a desirable outcome, but for understanding the fundamentals in depth, one should first focus on a setting that is more tractable.
>
> Additionally, please note that finding an equilibrium in general nonconvex-nonconcave minimax games is intractable [5]. As such, extra structural assumptions are needed to guarantee convergence to equilibrium for minimax optimization and multiplayer games in general. Our **quasi**-strong monotonicity and **star**-cocoercivity assumptions partly do that as they include classes of structured non-monotone games (see Appendix F), but handling even more relaxed structural assumptions would be an interesting future research direction. At the moment, however, it will require a distinct set of update rules and techniques, such as extragradient-type updates or multi-timescale step-sizes.
>
> [2] Stich. Local SGD converges fast and communicates little. ICLR, 2019.
>
> [3] Khaled et al. Tighter theory for local SGD … data. AISTATS, 2020.
>
> [4] Karimireddy et al. SCAFFOLD:  … federated learning. ICML, 2020.
>
> [5] Diakonikolas et al. Efficient Methods for Structured Nonconvex-Nonconcave Min-Max Optimization. AISTATS, 2021.
>
>
> ## Questions
>
>
> > 1. In classical FL, all the clients have a collaborative goal, and intuitively, by leveraging information from other clients, they would get a better result than training on their own…
>
>
> In the application scenarios that we have in mind, clients may have competing objectives but that does not necessarily imply that they are harming each other’s benefits. As a basic example, let us first recall the Generative Adversarial Networks (GANs), which is trained via solving the minimax game $\min_{G} \\, \max_{D} L(G,D)$ between the generator $G$ and the discriminator $D$. Here, the objective is designed to let $G$ and $D$ compete, but this lets both the generation capability of $G$ and the discriminative ability of $D$ improve.
>
>
> Now, as a similar but more timely application, consider a debate among multiple LLMs performed to reach an agreement, given a certain prompt. This can be modeled as searching for an equilibrium in a multiplayer game with not fully cooperative objectives (as different LLMs may initially output varying responses), but in the end, this leads to enhancement in the models’ reasoning quality or factual accuracy [6, 7]. This idea can be extended to joint training/fine-tuning of multiple language models, where the FL protocol (in particular, MpFL) would play an essential role, as centralized training requires jointly updating every model’s parameters at every iteration, which is unrealistic.
>
>
> [6] Du et al. Improving … Multiagent Debate, 2023.
>
>
> [7] Jacob et al. The Consensus Game: … Equilibrium Search. ICLR, 2024.
>
>
> > 2. typo: line 212: etragradient -> extragradient
>
>
> We thank the reviewer for pointing this out. We will update it in the revision.
>
>
> > 3. Figure 2(c) has a weird large shaded region in the bottom left.
>
>
> We thank the reviewer for taking a careful look. This is because we use the logarithmic scale for relative errors and one of the stochastic runs hit a very small loss by chance.
>
>
> > 4. Are all the experiments using a constant step size? Are there any numerical results to verify the sub-linear convergence to exact equilibrium in the stochastic case with decreasing step sizes?
>
>
> In the current version, we only included the experiments using a constant step-size to keep the setups simple, but internally, we also verified the sublinear convergence to the equilibrium using our step-size rule from Theorem 3.6. Unfortunately, we are not allowed to include any figures (or link to them) during the rebuttal this time, but in the revision, we will include an additional plot displaying those results.
>
> **We hope that our response has resolved the reviewer’s concerns. We believe that all points raised as weaknesses were mostly clarifications that can easily be handled in the camera-ready version of our work. If we have successfully addressed your concerns, please consider raising your mark. If you believe this is not the case, please let us know so that we have a chance to respond.**

---

> > ### Comment · Reviewer_jQKH · 2025-08-03
> > **The stochastic behavior of the experiment**
> >
> > Thanks for your response.
> >
> > Regarding Question 3, why does the small loss only occur between rounds 1000-3000? Does it imply that the numerical behavior of the algorithm is very unstable with large fluctuations?

---

> > > ### Author Response · Authors · 2025-08-04
> > >
> > > Unusually small or large losses (outliers) can occur in stochastic settings, and this is not specifically tied to rounds 1000–3000 or properties of PEARL-SGD—it can happen at any point during any algorithm's execution. **This does not indicate that PEARL-SGD is unstable, but rather results from the inherent randomness in the stochastic gradients.** Our theory predicts that PEARL-SGD converges in expectation to a neighborhood of the solution, where the radius of the neighborhood depends on the variance of the stochastic gradients and the value of $\tau$. The mean plot (the main line within the shaded region) confirms this prediction, and overall, the plot clearly shows a consistent pattern that $\mathbf{x}\_p$ tends to converge closer to $\mathbf{x}\_\star$ when $\tau$ is larger.
> > >
> > > Please also note that the y-axis of the plot is in *logarithmic scale*. The shaded region represents the standard deviation computed in the *original (linear) scale*, and due to this scale difference, it appears significantly distorted below the mean line. In contrast, the region above the mean line does not appear excessively large.
> > >
> > > **We hope that this clarification makes it clear that our PEARL-SGD algorithm is stable and behaves exactly as predicted by the theory, and that the plot is not evidence of instability.**

---

> ### Author Response · Authors · 2025-08-06
>
> As the discussion period is nearing its end, we would like to provide the reviewer jQKH a kind reminder and ask if they have any further questions.
>
> As noted in our previous rebuttals, we expect that our responses would have resolved the reviewer’s concerns. We believe that all points raised as weaknesses are simple clarifications that we can easily handle in the final revised version of our work. **If the reviewer agrees, please consider raising your mark to provide support. If this is not the case, we would appreciate the opportunity to resolve the reviewer’s remaining concerns.**

---

> > ### Comment · Reviewer_jQKH · 2025-08-07
> >
> > Thanks for your detailed response, I will raise my score.

---

> > > ### Author Response · Authors · 2025-08-08
> > >
> > > We thank the reviewer for the increased support and for finding our work interesting and well-written. We appreciate your support and the time you dedicated to reviewing our work.

---

### Official Review · Reviewer_3R9y · 2025-07-01

**Clarity:** 3
**Significance:** 3
**Originality:** 3
**Rating:** 4
**Confidence:** 3

**Summary:**

This paper proposes MpFL, a novel federated learning framework that models clients as rational game players with individualized objectives and seeks Nash equilibrium through collaborative, decentralized training under a central server. This paper provides thorough theoretical analysis and convergence proofs, along with supporting numerical experiments.

**Questions:**

(a) The paper claims in line 175 that PFL is a special case of the proposed MpFL. However, this statement is rather superficial, as it merely presents an example of a regularization-based approach in PFL for comparison. In fact, the standard formulation of PFL is $\min_{x_i} \frac{1}{m}\sum_i f_i(x_i) + \phi([x_i])$. Here, $\phi$ denotes a personalized regularization term, which does not necessarily have to be defined as a model consistency regularizer—it can take various other forms. Therefore, my question is: What are the essential differences between the proposed MpFL framework and this more general formulation of PFL where $\phi$ can be arbitrary?

(b) An important property in federated learning is the need to handle unexpected client dropouts caused by unstable connections or communication failures. As a result, most federated learning methods are designed to support partial client participation during training in order to address this practical issue. However, I did not find any discussion of this aspect in the MpFL framework.
Does the proposed MpFL method support partial client participation, and if so, how is it implemented?

(c) What are the practical use cases of the proposed framework? How does it fundamentally differ from the setting where $n$ independent players each participate in a game until reaching equilibrium?

**Ethical Concerns:**

["NO or VERY MINOR ethics concerns only"]

**Final Justification:**

I still maintain my rating as accept. Thank the authors for the submission.

**Limitations:**

This topic has already been discussed in the main text.

**Quality:**

3

**Strengths And Weaknesses:**

strength:

(a) The paper is well-motivated and offers a novel perspective distinct from traditional federated learning.

(b) The paper provides comprehensive theoretical analysis to demonstrate the effectiveness of the proposed algorithm.

weakness:

(a) Although the proposed approach is significantly different from traditional FL, the discussion regarding its relationship to PFL is insufficient and should be further emphasized. Essentially, the goal of the PFL framework is to train local optimal solutions with a common regularization constraint, where this regularization term is often carefully designed for different tasks. When facing multi-player game tasks, the proposed MpFL framework also appears to be a special case of the generalized PFL framework. The authors should include more discussion to clarify this connection.

(b) The numerical experiments are conducted on a relatively small scale. At present, it also seems difficult to distinguish the practical applications of this framework, and the differences between this framework and independent n-player games are not clearly demonstrated.

---

> ### Author Rebuttal · Authors · 2025-07-31
>
> We sincerely thank the reviewer for the positive evaluation, constructive feedback and their time and effort spent reviewing. Below, we provide responses to each point mentioned by the reviewer.
>
> ## Weaknesses
>
>
> > (a) Although the proposed approach is significantly different from traditional FL, the discussion regarding its relationship to PFL is insufficient and should be further emphasized…
>
>
> Please refer to our response to the Question (a) below. We thank the reviewer for the thoughtful question and suggestion.
>
>
> > (b) The numerical experiments are conducted on a relatively small scale. At present, it also seems difficult to distinguish the practical applications of this framework, and the differences between this framework and independent n-player games are not clearly demonstrated.
>
> In our proposed MpFL framework, all players still participate independently in the game to reach equilibrium. However, the important distinction with existing works on games is that the **proposed PEARL-SGD algorithm allows players to communicate less frequently.** In centralized training of games (which is presumably what the reviewer is referring to by “independent $n$-player games”), each player accesses other players’ actions at every iteration. This imposes a significant communication overhead, which PEARL-SGD aims to resolve by communicating only once in a while. We explain this in lines 95-103 and emphasize reduced communication as the central goal of MpFL, achieved by PEARL-SGD. Please also refer to our response to Question (c) below for the potential applications of MpFL, which further clarifies why these concepts are new and relevant.
>
> We agree with the reviewer that our experiments are relatively small-scale. **However, we believe that this should not be considered a significant weakness of our work, where we primarily claim conceptual and theoretical contributions. Not all papers should focus on large models or complex experimental settings.**  The primary goal of our work is to introduce this new theoretical framework (MpFL) and understand in depth the benefits and limitations of local methods for solving multiplayer games.
>
> ## Questions
>
> > (a) The paper claims in line 175 that PFL is a special case of the proposed MpFL. However, this statement is rather superficial, as it merely presents an example of a regularization-based approach in PFL for comparison…
>
>
> We only discuss the model consistency regularizer in the paper for simplicity. However, any PFL problem formulated in the form $\min\_{(x^1,\dots,x^n)} \frac{1}{n} \sum\_{i=1}^n h\_i (x^i) + \phi (x^1, \dots, x^n)$ can be reformulated as MpFL by setting $f\_i (x^1,\dots, x^n) = h\_i (x^i) + n \phi (x^1, \dots, x^n)$.
>
> On the other hand, there are some PFL formulations that do not follow the above structure. For instance, some influential papers [1, 2] proposed to keep a global model $w$ along with local models $x^i$, and such cases are not directly covered by the current formulation of MpFL.
>
>
> Therefore, we believe the correct viewpoint is that **PFL and MpFL are closely related setups but neither one is strictly more general than the other. However, the subclass of PFL using the formulation mentioned by the reviewer could be viewed as an instance of MpFL.** We will expand the current paragraph in lines 175-183 to include the above discussion. We thank the reviewer for pointing this out.
>
>
> [1] Fallah et al. Personalized Federated Learning with Theoretical Guarantees: A Model-Agnostic Meta-Learning Approach. NeurIPS, 2020.
>
>
> [2] Dinh et al. Personalized Federated Learning with Moreau Envelopes. NeurIPS, 2020.
>
>
> > (b) An important property in federated learning is the need to handle unexpected client dropouts caused by unstable connections or communication failures…
>
>
> In the case of client dropout, we can perform the essentially same analysis as in Section C.1 to obtain a convergence bound, with the net effect of probabilistic dropout being a reduction of effective step-size by the dropout probability. For the purpose of our paper, we intend to provide a clean, foundational analysis under an idealized setup and leave the extensions open for the future. However, we could include the above discussion and a simple sketch of ideas in the revision. We thank the reviewer for the important and relevant suggestion.
>
>
> > (c) What are the practical use cases of the proposed framework? How does it fundamentally differ from the setting where  independent players each participate in a game until reaching equilibrium?
>
>
> We foresee the following potential applications, although at the moment, actual implementation would need more exploration.
>
> - Autonomous Vehicle Coordination: This can be modeled as a game where vehicles’ objectives are to adjust driving strategies (speed, lane changes etc.) while adapting to the actions of others. Equilibrium is when each vehicle's strategy is optimal, given others’ strategies. FL will be needed in this scenario, as performing centralized training through real-time communication between individual players (vehicles) will be extremely challenging.
>
>
> - Multi-agent debate in language models: Given a prompt, LLMs can debate to reach agreement (equilibrium in a game) to enhance reasoning quality [3, 4]. This idea can be extended to joint training/fine-tuning of multiple language models. Here the FL setting would be the suitable theoretical description of the process, as opposed to the centralized training, which will require jointly updating every model’s parameters at every iteration (which is clearly unrealistic).
>
>
> Note that MpFL framework allows each player to perform multiple local updates based on other players’ actions given from the past (previous communication step), making it **suitable and relevant to practical scenarios where players are unlikely to have real-time access to others’ updated actions.**
>
>
> [3] Du et al. Improving … Multiagent Debate, 2023.
>
>
> [4] Jacob et al. The Consensus Game: … Equilibrium Search. ICLR, 2024.
>
> **We hope that our response has resolved the reviewer’s concerns. If we have successfully addressed your concerns, please consider raising your mark. If you believe this is not the case, please let us know so that we have a chance to respond.**

---

> > ### Comment · Reviewer_3R9y · 2025-08-02
> > **Responses to authors**
> >
> > I appreciate the authors’ detailed explanation. From my current understanding, MpFL actually also falls within the optimization paradigm of PFL (please correct me if I am mistaken), and is essentially a PFL optimization problem at its core. This does not diminish the contribution of MpFL; I just want to clarify this point. If the only difference lies in changing the task objective, I believe it is not necessary to elaborate excessively, defining an appropriate regularizer and local objective should be sufficient. If the authors could provide a more rigorous definition and proof that PFL problems can generally be viewed as a subset of MpFL problems, I would recommend adding a dedicated section to introduce this. This would help the community gain a better understanding of the topic.
> >
> > Moreover, conducting larger-scale experiments would further enhance the impact of the proposed method. The FL community typically focuses on large-scale scenarios, and the multi-agent tasks mentioned by the authors often involve even larger scales. Therefore, running simulation experiments on commonly encountered scales of several hundred or even thousands of nodes would be highly meaningful. I have no further questions and will maintain my current score.

---

> > > ### Author Response · Authors · 2025-08-08
> > >
> > > We sincerely thank the reviewer for their thoughtful feedback. In the revision, we will include a dedicated section that explains the connection to PFL in greater depth. We appreciate your positive evaluation of our conceptual and theoretical contributions, as well as the time you dedicated to reviewing our work.

---

### Official Review · Reviewer_1eaY · 2025-07-03

**Clarity:** 3
**Significance:** 2
**Originality:** 3
**Rating:** 4
**Confidence:** 3

**Summary:**

This paper studies a new federated learning (FL) paradigm inspired by a game-theoretic context and proposes PEARL-SGD. It includes theoretical analysis to support convergence guarantees under several scenarios. The authors verify the findings through empirical experiments.

**Questions:**

The questions are primarily included in the weaknesses part.

**Ethical Concerns:**

["NO or VERY MINOR ethics concerns only"]

**Final Justification:**

Based on the paper’s theoretical contribution, I raise my rating to a weak accept. Nevertheless, I still have reservations about the experimental section, as the current results make it difficult to fully convince readers of the method’s effectiveness.

**Limitations:**

Yes

**Quality:**

3

**Strengths And Weaknesses:**

Strengths:
* This paper is well-structured and easy to follow.
* The theortical contribution is solid, with several scenarios. The results presented in the theorems are convincing.

Weaknesses:
* One of the major concerns with the proposed method is the communication cost. Since the master node distributes $\mathbf{x}$ back to the players at each communication step $r$, the communication overhead for $\mathbf{x}$ becomes $n$ times higher compared to traditional FL methods. This significantly reduces the practicality of the proposed approach when applied to large-scale models.
* The comparison with existing methods is limited, and the experiments are conducted on relatively small-scale models. Since the goal of the proposed PEARL-SGD is still to obtain a well-performing global model, comparisons with methods like FedAvg, SGDA, and other standard FL approaches should be feasible.
* The theoretical resulst of this paper relies on several assumptions. Some of them are quite common and widely used in FL papers, but some of them are less common in FL research such as Quasi-strong monotonicity and Star-cocoercivity. The authors should provide more discussion on the validity and applicability of these assumptions in practical FL settings.
* I understand that the game-theoretic context may rely on convexity assumptions for theoretical analysis, but it would be helpful if the authors could explain why the non-convex setting is not considered or applicable.

---

> ### Author Rebuttal · Authors · 2025-07-31
>
> We sincerely thank the reviewer for viewing our paper as interesting and well-written, for constructive feedback, and for their time and effort spent reviewing. Below we provide responses to each point.
>
> > One of the major concerns with the proposed method is the communication cost. Since the master node distributes $\mathbf{x}$ back to the players at each communication step $r$, the communication overhead for $\mathbf{x}$ becomes $n$ times higher compared to traditional FL methods. This significantly reduces the practicality of the proposed approach when applied to large-scale models.
>
> Yes, in our proposed algorithm PEARL-SGD, as we also discussed lines 192-202 of our work, the master node distributed the vector $\mathbf{x} = (x^1, \dots, x^n)$ to all players. In practical scenarios, this will indeed make our method particularly suitable for cross-silo FL setups with a relatively small number of organizations/players involved. That said, let us highlight here that the need for collecting all strategies is present in virtually any prior work considering algorithms for multiplayer games, distributed or not (see references in lines 1038-1048), as each player has to know all the others’ strategies/models. **Therefore, this is not the limitation specific to the MpFL framework. Rather, MpFL partially addresses this issue by communicating less frequently, as opposed to the centralized games,** where all strategies have to be communicated at every step. Using local steps (via PEARL-SGD), we are able to reduce the communication cost.
>
> Regarding the practicality of our results, **both the PEARL-SGD algorithm and the framework of MpFL we propose could be used in any prior practical applications of multiplayer games** to reduce communication complexity (via local updates).
>
> > The comparison with existing methods is limited, and the experiments are conducted on relatively small-scale models. Since the goal of the proposed PEARL-SGD is still to obtain a well-performing global model, comparisons with methods like FedAvg, SGDA, and other standard FL approaches should be feasible.
>
> The goal of our proposed MpFL and PEARL-SGD is to reach equilibrium while **each player is carrying their own model (action)** and objective function to minimize, which is very different from the standard FL setting where obtaining a good global model is sufficient. In the MpFL setup, the dimensionality and semantics of each player’s action may vary (e.g., if they are neural network parameters, the local models’ architectures and model sizes could be different). This prevents one from naively applying the methods such as FedAvg or (Local) SGDA, which require averaging the local models. We provide a detailed explanation on these points in lines 149-174 and in Appendix B. Additionally, in Appendix B, we provide a simulation result on what happens if one simply ignores the conceptual mismatch and nonetheless applies FedAvg to the MpFL setup, showing that FedAvg diverges even for a simple quadratic problem.
>
> We agree with the reviewer that our experiments are relatively small-scale. **However, we believe that this should not be considered a significant weakness of our work, where we primarily claim conceptual and theoretical contributions. Not all papers should focus on large models or complex experimental settings.**  The primary goal of our work is to introduce this new theoretical framework (MpFL) and understand in depth the benefits and limitations of local methods for solving multiplayer games.
>
> > The theoretical result of this paper relies on several assumptions. Some of them are quite common and widely used in FL papers, but some of them are less common in FL research such as Quasi-strong monotonicity and Star-cocoercivity. The authors should provide more discussion on the validity and applicability of these assumptions in practical FL settings.
>
> We thank the reviewer for the suggestion. To that end, **please let us point the reviewer to “Appendix F: Discussion on Theoretical Assumptions” in the paper, where we provide the exact details the reviewer requested.** We would be happy to move parts of this discussion to the main paper, once we are given additional space.
>
> Our assumptions are standard conditions used in many recent papers in multiplayer games (see lines 203-219). Let us also highlight here that our assumptions of *quasi*-strong monotonicity and *star*-cocoercivity include classes of structured non-monotone games, where the players’ objective functions can be nonconvex [1, Appendix A.6].
>
> [1] Loizou et al. Stochastic Gradient Descent-Ascent and Consensus Optimization for Smooth Games: Convergence Analysis under Expected Co-coercivity. NeurIPS, 2021.
>
>
> > I understand that the game-theoretic context may rely on convexity assumptions for theoretical analysis, but it would be helpful if the authors could explain why the non-convex setting is not considered or applicable.
>
> Indeed, convergence analysis in the game theoretic setup typically requires convexity or similar assumptions. As explained in [2], finding an equilibrium in general nonconvex-nonconcave minimax games is intractable. As such, extra structural assumptions are needed to guarantee convergence to equilibrium for minimax optimization and multiplayer games in general.
>
> Assumptions in our work are in line with this literature — as explained above, *quasi*-strong monotonicity and *star*-cocoercivity include classes of structured non-monotone games (generalizing the more commonly used assumptions of strong monotonicity and cocoercivity). Handling even more relaxed structural assumptions is an interesting future research direction, but it will require different mechanisms, such as two-timescale step-sizes and extragradient-type updates.
>
> [2] Diakonikolas et al. Efficient Methods for Structured Nonconvex-Nonconcave Min-Max Optimization. AISTATS, 2021.
>
> **We hope that our response has resolved the reviewer’s concerns. We believe that all points raised as weaknesses were mostly clarifications that can easily be handled in the camera-ready version of our work. If we have successfully addressed your concerns, please consider raising your mark. If you believe this is not the case, please let us know so that we have a chance to respond.**

---

> > ### Comment · Reviewer_1eaY · 2025-08-05
> >
> > Thanks the authors for the rebuttal.
> >
> > My main concerns is still with the experimental evaluation.
> >
> > * First, a minor point: while I fully agree that “not all papers should focus on large models or complex experimental settings,” I don’t think this alone justifies the lack of experiments. In my view, a stronger response would either highlight specific technical challenges in conducting larger-scale experiments or cite relevant prior work to demonstrate that focusing on small-scale examples is sufficient and accepted in the field.
> > * Second, I am not suggesting to "applies FedAvg to the MpFL setup", my point is that "comparisons with methods like FedAvg, SGDA, and other standard FL approaches should be feasible".  As this work proposes an FL method, it would be more convincing if it were compared against widely-used or basic methods. Otherwise,  it risks remaining just a strategy on paper.  I understand that in the MpFL setting, “the dimensionality and semantics of each player’s action may vary,” but the overarching goal remains to train a well-performing global model. Given that, why not consider a simplified setting where all players share the same architecture and compare MpFL against FedAvg? At the very least, such a comparison could provide insight into how the equilibrium-based loss design performs relative to conventional FL objectives.
> >
> > Overall, I appreciate the design of MpFL and recognize the theoretical contributions, it’s an interesting and thoughtful method. However, I hope it can go beyond being just a conceptual and theoretical illustration and be verified as a solid method through empirical experiments.

---

> > > ### Author Response · Authors · 2025-08-07
> > >
> > > We greatly thank the reviewer for engaging in the discussion and for acknowledging our MpFL and theoretical contributions as interesting and thoughtful.
> > > We will address the second point first, as we believe it is more crucial.
> > >
> > > > Second, I am not suggesting to "applies FedAvg to the MpFL setup", my point is that "comparisons with methods like FedAvg, SGDA, and other standard FL approaches should be feasible" …
> > >
> > >
> > > We would like to respectfully clarify once more that **MpFL is not an FL framework aimed at training a global model**, as we explain in our main paper, in Appendix B, and in our previous response. In MpFL, *each player optimizes their own local model $x^i$*, based on their individual objectives and interactions with others. In this setting, there is no notion of a shared global model in the optimization goal. Due to this distinction—a conceptual mismatch, PEARL-SGD cannot be simply compared against FedAvg on ordinary FL tasks as the reviewer believes. **Our MpFL and PEARL-SGD are not designed to improve upon basic methods in terms of the metrics, but to introduce a totally new problem formulation which classical FL framework and algorithms (such as FedAvg) fail to capture and handle.** This is what we demonstrate in Appendix B through a carefully designed synthetic experiment.
> > >
> > >
> > > While we understand the reviewer’s willingness to see more comparisons with “widely used or basic methods”, we emphasize that **this would be analogous to asking a GAN** (which jointly optimizes a generator and a discriminator with fundamentally distinct roles), trained by solving the minimax problem of the form $\min\_G \max\_D L(G,D)$, **to be compared against a standard classifier trained by minimizing the cross-entropy loss.** Such comparisons, if not impossible, would not be informative, as the objectives, assumptions, and intended outcomes are all misaligned.
> > >
> > >
> > > > First, a minor point: while I fully agree that “not all papers should focus on large models or complex experimental settings,” I don’t think this alone justifies the lack of experiments. In my view, a stronger response would either highlight specific technical challenges in conducting larger-scale experiments or cite relevant prior work to demonstrate that focusing on small-scale examples is sufficient and accepted in the field.
> > >
> > >
> > > We initially chose to keep our response concise and avoid lengthy elaboration. However, as the reviewer correctly points out, it is worth emphasizing that several foundational works in federated learning [1, 2, 3, 4] have similarly prioritized theoretical development and provided only small-scale simulations, often on regularized logistic regression problems where assumptions such as strong convexity and smoothness hold. **We believe our paper follows a similar structure: presenting a concrete theoretical framework, accompanied by basic numerical experiments that serve as a proof-of-concept. In this respect, we do not view our contribution as any weaker than those prior works.**
> > >
> > > Additionally, we would like to note that scaling up experiments to involve multiple neural network models in a multiplayer game setting would result in significantly greater computational costs. This is due to the nature of MpFL, where all players’ model information is exchanged at communication steps. While such large-scale experiments are certainly feasible—especially in the context of specific applications supported by organizational resources—our current work is intended as a theoretical foundation. We view the implementation of more extensive empirical studies as an important direction for future work, focusing on domain-specific applications of the MpFL framework.
> > >
> > > [1] Stich. Local SGD converges fast and communicates little. ICLR, 2019.
> > >
> > > [2] Khaled et al. Tighter Theory for Local SGD on Identical and Heterogeneous Data. AISTATS, 2020.
> > >
> > > [3] Gorbunov et al. Local SGD: Unified theory and new efficient methods. AISTATS, 2021.
> > >
> > > [4] Mishchenko et al. ProxSkip: Yes! Local Gradient Steps Provably Lead to Communication Acceleration! Finally! ICML, 2022.
> > >
> > > *We hope that our follow-up response has addressed the reviewer’s concerns. We believe that the back-and-forth discussion has helped identify points that, once properly explained and clarified, will strengthen the paper.
> > > If the reviewer agrees, we kindly ask you to consider raising your score in support. If not, we would appreciate the opportunity to address any remaining concerns.*

---

> > > > ### Comment · Reviewer_1eaY · 2025-08-09
> > > >
> > > > Thank you to the authors for the detailed response.
> > > >
> > > > Based on the paper’s theoretical contribution, I will raise my rating to a weak accept. Nevertheless, I still have reservations about the experimental section, as the current results make it difficult to fully convince readers of the method’s effectiveness.

---

### Official Review · Reviewer_v3ed · 2025-07-03

**Clarity:** 3
**Significance:** 4
**Originality:** 4
**Rating:** 5
**Confidence:** 4

**Summary:**

This paper introduces Multiplayer Federated Learning (MpFL), a game-theoretic framework for federated learning where clients are viewed as strategic players with possibly misaligned objectives. The authors propose a communication-efficient algorithm, PEARL-SGD, and prove convergence guarantees in both deterministic and stochastic regimes under standard game-theoretic assumptions (QSM, SCO). The algorithm is validated on synthetic and control-theoretic examples.

**Questions:**

1. In section 2.2, the authors mentioned the relation between MpFL and federated minimax problem, where the later seems to be a special case in the MpFL framework (2-player). Does it mean that in federated minimax problems, the players are different from the clients (computing agents)?


2. Since bilevel optimization is essentially a two-player (leader-follower) Stackelberg game, what is the relation between MpFL and federated bilevel optimization (FBO) [1,2,3] problems? Is FBO a special case of MpFL, just like federated minimax problem?

[1] Tarzanagh et al. FedNest: Federated Bilevel, Minimax, and Compositional Optimization. ICML 2022.

[2] Qiu et al. Zeroth-Order Methods for Nondifferentiable, Nonconvex, and Hierarchical Federated Optimization. NeurIPS 2023.

[3] Yang et al. First-Order Federated Bilevel Learning. AAAI 2025.


3. As I mensioned in the Weaknesses 1, the lack of numerical comparison would weaken the paper. Is it possible to add some comparison under special cases (eg. 2-player minimax problem)? Can MpFL address them as well? I believe adding comparisons would strengthen the numerics significantly.

**Ethical Concerns:**

["NO or VERY MINOR ethics concerns only"]

**Final Justification:**

The framework this work proposes is novel and interesting. I will maintain my rating as accept.

**Limitations:**

Yes, discussed.

**Paper Formatting Concerns:**

No.

**Quality:**

3

**Strengths And Weaknesses:**

### Strengths:

1. MpFL addresses a gap in the FL literature by modeling non-cooperative clients via an n-player game formulation. It generalizes classical FL and personalized FL under a rigorous game-theoretic framework.

2. The paper provides linear convergence to a neighborhood under constant step-size and exact convergence under decreasing step-size, improving over classical local SGD in heterogeneity settings.

3. Quantifies how larger communication intervals (τ) can lead to reduced communication without compromising performance.

4. The heatmap visualization is insightful, showing the trade-off between step-size and communication frequency.

### Weaknesses

1. The experiments are too synthetic and not representative of typical federated learning applications. No comparison to recent FL methods that incorporate strategic or incentive-aware behavior (e.g., [10], [29]).

2. Assumptions might be restrictive, some discussion or justification on when they hold in practice would help.

3. The authors admit the method suits small-n, cross-silo setups, but don’t address general scalability. More discussion is needed.

---

> ### Author Rebuttal · Authors · 2025-07-31
>
> We sincerely thank the reviewer for the positive evaluation, constructive feedback, and their time and effort spent reviewing. We are delighted to see that the reviewer views our work as correctly addressing a missing part in the FL literature. Below, we provide responses to each point mentioned by the reviewer.
>
> ## Weaknesses
>
> > 1. The experiments are too synthetic and not representative of typical federated learning applications. No comparison to recent FL methods that incorporate strategic or incentive-aware behavior (e.g., [10], [29]).
>
> Our experiment setup has been considered very often in the multiplayer games literature [1, 2] and minimax optimization community [3, 4]. While it is synthetic, it is a standard testbed used in the literature and it successfully highlights the communication gain attained by PEARL-SGD.
>
> Regarding the methods considering strategic and incentive-aware clients, including [5, 6] (respectively cited as [10, 29] in the paper), we would like to refer the reviewer our related paragraph in Appendix A (lines 1060-1069), discussing why direct experimental comparison with them was not adequate for this paper. We believe that the connection is interesting and MpFL may potentially incorporate those prior work, but at the moment, our focus — communication-efficient equilibrium search in general games — is largely different from theirs.
>
> [1] Salehisadaghiani & Pavel. Distributed Nash equilibrium seeking: A gossip-based algorithm. Automatica, 2016.
>
> [2] Ye & Hu. Distributed Nash equilibrium seeking by a consensus based approach. IEEE Transactions on Automatic Control, 2017.
>
> [3] Mokhtari et al. A Unified Analysis of Extra-gradient and Optimistic Gradient Methods for Saddle Point Problems: Proximal Point Approach. AISTATS, 2020.
>
> [4] Beznosikov et al. Stochastic gradient descent-ascent: Unified theory and new efficient methods. AISTATS, 2023.
>
> [5] Blum et al. One for one, or all for all: Equilibria and optimality of collaboration in federated learning. ICML 2022.
>
> [6] Dorner et al. Incentivizing Honesty among Competitors in Collaborative Learning and Optimization. NeurIPS, 2023.
>
> > 2. Assumptions might be restrictive, some discussion or justification on when they hold in practice would help.
>
> We do work on a specific set of assumptions, which are standard in the literature on stochastic gradient descent-ascent type methods. These assumptions are not intended to be directly applied to practical models (neural networks). Our analysis parallels how numerous theoretical works in classical FL have been done [7, 8, 9], assuming strong convexity while primary applications are ML models. Please note that our primary goal is to present a novel framework under an idealized setup (understanding the simple setting first will allow the community to build on strong foundations).
>
> [7] Stich. Local SGD converges fast and communicates little. ICLR, 2019.
>
> [8] Khaled et al. Tighter theory for local SGD … data. AISTATS, 2020.
>
> [9] Karimireddy et al. SCAFFOLD:  … federated learning. ICML, 2020.
>
> > 3. The authors admit the method suits small-n, cross-silo setups, but don’t address general scalability. More discussion is needed.
>
> As we mentioned in lines 192-202, the communication cost will grow linearly with the number of players $n$, which limits the scalability of MpFL in its basic form. However, (as we also discuss in lines 192-202) we believe that this could be resolved using techniques such as gradient compression or random sampling of the players performing the local updates.
>
> ## Questions
>
> > 1. In section 2.2, the authors mentioned the relation between MpFL and federated minimax problem, where the latter seems to be a special case in the MpFL framework (2-player). Does it mean that in federated minimax problems, the players are different from the clients (computing agents)?
>
> In MpFL, the clients of FL act as individual players of the game, while in federated minimax optimization (FMO) each client has access to both (min and max) players’ objective functions and their actions/strategies. So **yes, in FMO, players and clients are distinct concepts, but in MpFL, clients are players, aiming to minimize their own local objectives by adjusting their actions.**
>
> > 2. Since bilevel optimization is essentially a two-player (leader-follower) Stackelberg game, what is the relation between MpFL and federated bilevel optimization (FBO) problems? Is FBO a special case of MpFL, just like federated minimax problem?
>
>
> Indeed, bilevel optimization is a two-player game with hierarchical structure. Therefore, MpFL is closely related to federated bilevel optimization (FBO) as well, similarly to FMO. In particular, taking inspiration from FBO, we believe that MpFL for hierarchical games (where the order of players is important) is an interesting future direction. We could easily add a discussion on these connections in the revision of our work. We thank the reviewer for the suggestion.
>
> > 3. As I mentioned in the Weaknesses 1, the lack of numerical comparison would weaken the paper. Is it possible to add some comparison under special cases (eg. 2-player minimax problem)? Can MpFL address them as well? I believe adding comparisons would strengthen the numerics significantly.
>
> In Appendix B, we provide a simple simulation result on what happens if one simply ignores the conceptual mismatch and nonetheless applies FedAvg to the MpFL setup — this shows that FedAvg diverges even for a simple quadratic problem, hence highlighting that PEARL-SGD is needed. In the revision, we will add a similar demonstration for the two-player case as well. We thank the reviewer for the suggestion.

---

> > ### Comment · Reviewer_v3ed · 2025-08-05
> >
> > I thank the authors for the response. The framework this work proposes is novel and interesting. I will maintain my rating as accept.

---

> > > ### Author Response · Authors · 2025-08-08
> > >
> > > We sincerely thank the reviewer for their thoughtful feedback and positive evaluation. We appreciate your support and the time you dedicated to reviewing our work.

---

### Note · Authors · 2025-08-16

We sincerely thank all reviewers for their time, effort, and constructive suggestions.
We will incorporate the feedback into the revised version of the paper by further clarifying connections to relevant fields (e.g., Personalized FL, federated bilevel optimization) and expanding important discussions.

Overall, we are delighted that the reviewers found *our framework and results to be well-motivated, novel, and interesting, and our theoretical contributions solid and convincing.* We view our work as a multiplayer-game analogue of established theoretical results in classical FL, focusing on convergence guarantees and communication efficiency under suitable assumptions, while additionally introducing a new conceptual framework of MpFL and a novel PEARL-SGD algorithm.

We would be grateful if the AC and reviewers recognize that, as a theoretical foundation, our work not only stands on equal footing with the already-published prior literature but also offers a new and meaningful direction for the FL literature.
Once again, we thank the AC and reviewers for their thoughtful consideration.

Best regards,

Authors

---

### Decision · Program_Chairs · 2025-09-17

**Decision:**

Accept (poster)

**Comment:**

This paper introduces Multiplayer Federated Learning (MpFL), a framework that models clients in federated learning as self-interested players with potentially misaligned objectives, analyzed through a game-theoretic lens. The authors propose an algorithm where each client performs independent local updates and communicates periodically, and show its convergence to a neighborhood of equilibrium with reduced communication. Theoretical results are supported by numerical experiments validating the approach. Even though the scale of the experiments is small, I agree with the reviewers that the theoretical contribution of the paper is significant enough to warrant acceptance.